

# Spatial estimation of air PM$_{2.5}$ emissions using activity data, local emission factors and land cover derived from satellite imagery

Hezron P. Gibe[1], Mylene G. Cayetano[1]

[1]Institute of Environmental Science, University of the Philippines, Diliman, Quezon City, Philippines

*Correspondence to*: Mylene G. Cayetano (mcayetano@iesm.upd.edu.ph)

**Abstract.** Exposure to air particulate matter (APM) is a presently relevant issue that affects the environment and the health of residents of many urban areas globally. In the Philippines, most existing studies and emission inventories have mainly focused on point and mobile sources, while research involving personal exposures to particulate pollutants is mostly lacking. This paper presents a method for estimating the amount fine (PM$_{2.5}$) particulate emissions in a test study site in Cabanatuan

City, Nueva Ecija in the Philippines by utilizing local emission factors, regionally procured data and land cover/land use (activity data) interpreted from satellite imagery. The estimated emissions have been mapped using geographic information systems (GIS) software. Results suggest that vehicular emissions from motorcycles and tricycles, as well as biomass-based household fuels (charcoal) and burning of agricultural waste largely contribute to PM$_{2.5}$ emissions in Cabanatuan City. Overall, the method used in this study can be applied to any study site, as long as on-site specific activity data, emission

factor and satellite-imaged land cover are available.

**1 Introduction**

Exposure to air particulate matter, especially fine particles smaller than 2.5 micrometers in size (PM$_{2.5}$), impacts visibility in the form of haze phenomena, can affect local and regional air quality, and can have lasting effects on climate. Many urban dwellers are at risk of high pollutant exposure, living in areas of high outdoor ambient pollution, and in many cities, the amount of pollution frequently exceeds WHO guideline values for air pollutants (Mage, *et al.*, 1996).The presence of PM$_{2.5}$

among other air pollutants urban cities in general represents a significantly high health risk for residents, as it is linked to increased morbidity and mortality risk, especially in incidences of various cardio-pulmonary diseases (Chen, *et al.*, 2008; Lin, *et al.*, 2016; Wu, *et al.*, 2013), birth defects (Goto, *et al.*, 2016), and cancer (Cassidy, *et al.*, 2007). PM$_{2.5}$ pollution is



also considered carcinogenic especially at the finest fractions (ultrafine particles) (Bocchi, *et al.,* 2016). This can be attributed to fine particulates acting as carriers of mutagenic and genotoxic compounds (Chen, *et al.,* 2016).

Sources of fine particulates come from many activities. A common source of $PM_{2.5}$, in urban areas comes from mobile

sources, directly emitted by internal combustion processes inside vehicles of all types (Andrade, et al., 2012; Ahanchian and Biona, 2014; Chen, et al., 2016).

In most of the reports from the Philippines, percentages of emissions from mobile sources are reported in inventories and using emission factors from non-local engine sources (CORINAIR, AP 42). In areas where urban and rural land uses are

both present, however, especially in the Philippine context, $PM_{2.5}$ emissions can be accounted for by other factors as well, such as particulates generated by the burning of biomass sourced from agricultural waste (Sarigiannis, et al., 2014; Kim Oanh, et al., 2011; Gadde, et al., 2009)

At present, the current basis for most measures for air quality monitoring and management are based on $PM_{10}$ and total

suspended particles (TSP) as an indicator. Standards for $PM_{2.5}$ in comparison have not been fully developed and implemented in small cities; emission inventories in general have likewise not been conducted in many cities and regional centers as well. Aside from this, such studies are often conducted every few years, if at all, presenting a lack of temporally resolved long-term historical data for air quality monitoring purposes.

This paper presents a spatial method for the estimation of air $PM_{2.5}$, by utilizing local emission factors, as well as satellite imagery and "activity data" from their interpretation (with the use of GIS software for mapping). The usage of the term "activity data" stems from the current process of the air pollution emissions inventory in most cities in the Philippines. The current method involves the location and identification of all emission sources in a given city, taking into account the type of emission (point, area, mobile), and nature of activity which produces the emissions. In this study, "activity data" is defined

as not only the type of air pollution generating activity, but also includes factors such as local population, density of households, density of emission-generating events, and the type and amount of various fuels used. This, in conjunction with various local emission factors, will be used to estimate total $PM_{2.5}$ emissions. A limitation of this study is that all emission sources are treated as being area sources, as this is required for the mapping process.

From the resulting maps, the study aims to determine areas of high concentration of $PM_{2.5}$, in total and by individual $PM_{2.5}$ sources. This method is specifically meant to explore this method for use in relatively small regional urban centers and cities in the Philippines; especially due to these cities being situated in locations where their land uses tend to be more diverse and more influenced by conventionally "rural" activities such as agriculture and the usage of traditional household fuels such as charcoal.





Another application for this study involves its potential use as planning aids for local governments; as this method can also be used in emission inventories for small cities. As such, the methods used in this study were developed for use by relevant personnel with minimal required training, in order to serve as a possible method of expediting the process of gathering
emission inventories.

## 2 Materials and methods

### 2.1 Gridded study area and land cover from satellite imagery

The test study was conducted in Cabanatuan City, Philippines. (Fig. 1) It is the former capital and largest city of the province of Nueva Ecija, with a land area of 190.67 square kilometers and an estimated population of 296,584 as of 2012. On average,
the population density is around 1,516 persons per square kilometer. The urban and rural population take up around half of the total population of the city each (Cabanatuan City SEP, 2015).

A 2.4 by 4.0 kilometer bounded section corresponding to the city center and its immediate environs was selected as the main study area. The town proper, (locally known as the *poblacion*) is highlighted in the map of the study area as shown in Fig. 2.
The boundaries of *barangays* (the smallest administrative division of a local government, a similar concept to town wards or districts) are marked in grey, and the constituent barangays of the *poblacion* are marked distinctly as a point of reference. This area represents much of the different land uses present within the city center and its surroundings; these include residential, commercial, and even agricultural areas within the short space of less than two kilometers from the national road. The study area, however, does not take into account a relatively sizeable commercial zone south of this enclosed region, as
well as the main industrial district of Cabanatuan City located near its eastern border, which is located around 10 kilometers east of the city center.

This area was mapped out in a 24 x 40 cell grid of 100 x 100 meter (1 ha / 0.01 km$^2$) cells. For each cell, the type of activity was interpreted from satellite images taken from Google Earth software. Detailed images from the ground level taken by
Google Street View (examples are shown in Fig. 3) were also used to verify building types (residential/commercial) when available. Satellite images were dated March 3$^{rd}$, 2016, while Street View images were dated September 2015. Additionally, maps of the same location from OpenStreetMap were also used as a reference for landmarks not present in the information in Google Earth, or present more updated information compared to ground features from the Street View images.

The choice to use Google Earth images as opposed to raw image data from sources such as Landsat images was made for several reasons. The method used in this study is intended to be used by personnel not necessarily familiar with or are trained in the processing of satellite imagery data, in which case, the researchers have opted to use Google Earth imagery as a





possible source in this manner. Images (in this area, sourced from Landsat and the European Space Agency (ESA)'s Copernicus program since 2015) have been processed and corrected for aberrations of the camera taking said satellite image, as well as collaged to only show higher quality images with minimal cloud cover. However, there are issues with these images that are worth mentioning. For one, the resulting image collage is not necessarily representative of the most current

features on the ground. Google Earth imagery is projected orthographically; while this allows for relatively accurate measurements of distance in the small scale, it can also result in the image location being deviated from its actual geographic coordinates. While the images themselves aren't georeferenced, the coordinates provided by the software are accurate enough to represent the actual surface without much deviation. Other issues include the limitations of accessing the metadata of the original images. In spite of these issues, these materials are sufficient for the uninitiated and can be very useful

regardless considering their intended purpose. It is very important, then, that other tools such as the Google Street View images or community-based initiatives for mapping such as OpenStreetMap be used. The usage of supporting documents such as existing local government land use plans and land cover maps, as well as actual verification of features at the ground level (ground truth, that is, information on surface features in the study area), is necessary, and was used in this study to verify land cover and land use features at the surface level.

Land cover features for each 100 x 100 m cell for areas inside the study site were identified and then mapped. These are classified by their possible land uses (i.e. residential/commercial, agricultural areas, roads, other surface features) and are associated with an emission type, to be used for estimating $PM_{2.5}$ levels later in the process. As seen in the map in Fig. 4, residential areas (represented in the map as "residential" cells, which account for households using liquefied petroleum gas

(LPG) as fuel) are spread widely throughout the study site, with a noticeable commercial district (represented by cells marked with "commercial") at its very center. At the same time, the cells immediately outside and some even inside the overall bound of the residential areas are categorized as either open space (a term used for characterizing areas that are not considered built-up), or agricultural areas. This is especially noticeable in the northwest and the southeast portions of the map; the northwest section is mostly agricultural, occupied by small households which are likely using fuels, and located

adjacent to the Pampanga River. In the southeast portion of the map, a residential area is seen next to cells with open space and agricultural areas; this represents recently built-up areas used for new residential developments that have been constructed in the past few years in Cabanatuan City.

Note that some cells are marked as land uses by themselves as they are currently considered to be "special" land uses, and

the emission factors for these cells are currently being studied: these are the "cemetery" and "terminal" cells, the latter corresponding to the central transport terminal of Cabanatuan City, currently considered a commercial area with a large concentration of vehicular emissions.





**2.2 Activity data and emission estimation**

All calculations that have been used to estimate emissions is based on a general formula used by the US EPA in the AP 42 Compilation of Air Pollutant Emission Factors (EPA, 1995), as shown in Eq. (1)

$$E = \text{A} \times \text{EF} \times \left(1 - \frac{\text{ER}}{100}\right), \tag{1}$$

5   where: E is equal to emissions, A is the "activity rate" (term used by EPA; this study uses "activity data" to describe this and other relevant factors pertaining to the quantity of fuel used and percentage of households using fuel), EF represents the emission factor (in this case, for PM$_{2.5}$), and ER is the overall emission reduction factor/efficiency in percent, if applicable. In this study, ER was also used to refer to other factors affecting the total amount of PM$_{2.5}$ emissions (such as factors not directly accounting towards the quantity of fuel used; ER factors also incorporate the activity of those using quantities of fuel

10   lower than average). For this study, factors used as "activity data" (A) and "emission reduction" (ER) will be collectively be referred to as emission estimation factors (EEF).

Emission factors for fuelwood, charcoal, vehicular emissions, and rice straw burning, were sourced from various local studies and projects. The EEF for each emission type is calculated depending on which metrics are relevant for each source

15   of PM$_{2.5}$. Household and population data were obtained from local government documents, particularly the Comprehensive Land Use Plan(s) and Socio-Economic Profile(s) of Cabanatuan City; information on total amount of fuel used by household is obtained from the national Household Electricity Consumption Survey (HECS), conducted in 2005 and 2011. Table 1 compiles the sources of activity data used in this study, in various units such as fuel consumption, population and household data, and agricultural land use data per year.

**Table 1.** Data sources for activity data, emission factors, and ER factors

| | |
|---|---|
| **Population data, land use** | 2016 (provisional) Cabanatuan CLUP Cabanatuan City SEP (2015) |
| **Activity data for household fuels; LPG, charcoal consumption** | 2011/2005 Household Energy Consumption Survey (HECS) Ground surveys |
| **Emission factors (PM$_{2.5}$) for fuelwood, charcoal** | Cayetano and Lamorena (2014) |
| **Activity data for PUVs and motorcycles, tricycles (MC/TC),** | Local government documents, Land Transportation Office (LTO) annual reports, ground surveys |



| | |
|---|---|
| **Emission factors for PUVs and MC/TCs** | In-house data |
| **Data on rice production and rice land agricultural area** | Cabanatuan City SEP (2015) |
| | 2016 (provisional) Cabanatuan CLUP |
| **Data on rice straw generated per amount rice produced** | Bakker, et al. (2013) |
| **Emission factor for rice straw burning** | In-house data |

Emissions for household fuel (charcoal) were estimated with the formula shown in Eq. (2):

$$E_{fuels} = (N_h \times HF) \times Q_{fuel} \times EF \times 0.01 \; , \tag{2}$$

where: $E_{fuels}$ is equal to emissions generated by charcoal fuels, $N_h$ is the estimated number of households (generated from city government data), and HF is the factor (in percent) of all households using charcoal as fuel, obtained from the HECS. $Q_{fuel}$ is the quantity of fuel in kilograms used per year by each household, sourced from the HECS and verified using sensitivity analysis by ground surveys (see section 2.3). EF corresponds to the emission factor for charcoal fuel $PM_{2.5}$ per year per square kilometer; this is then multiplied by 0.01 to scale to each 0.01 $km^2$ cell.

Emissions for motorcycles and tricycles were estimated with the formula shown in Eq. (3):

$$E_{vehicles} = (N_u \times DF \times NAF) \times (EF \times KT \times SDF) \times 0.01 \; , \tag{3}$$

where: $E_{vehicles}$ is equal to emissions generated by vehicles (motorcycles and tricycles, PUVs), $N_u$ is the estimated number of vehicle units (by type: MC/TCs, PUVs); multiplied by density factor DF corresponding to amount of vehicles per $km^2$ area in
the city and non-association (vehicles) factor NAF, which corresponds to an additional multiplier to the overall number of vehicles taking into account unregistered vehicles (not registered by the city, or are from outside Cabanatuan City). The DF and NAF were sourced and derived from city government data and verified using sensitivity analysis by ground surveys as well. EF corresponds to the emission factor for motorcycle and tricycle/PUV $PM_{2.5}$ per year per square kilometer per kilometer traveled. The emission factor is scaled to the average distance traveled by any given vehicle unit, here represented
as factor KT (kilometers traveled). Similar to the previous example, also multiplied by 0.01 to scale to each 0.01 $km^2$ cell.

Emissions for rice straw burning in agricultural areas were estimated with the formula shown in Eq. (4):

$$E_{straw} = \left(\frac{RS}{RA}\right) \times EF \times SF \; , \tag{4}$$

where: $E_{straw}$ is equal to emissions generated by rice straw burning, RS is the amount of rice straw produced per year, divided
by RA, which is the total area in hectares (0.01 $km^2$) used for growing of rice. EF is the in-house obtained emission factor for



rice straw burning $PM_{2.5}$ per year per square kilometer. SF is the survey factor, acting as the reduction factor from the study of Launio, et al. (2013); this represents the percentage of farming area where burning of rice straw as agricultural waste is used.

After the estimated emissions for each cell have been calculated, they were mapped using ArcMap (ArcGIS 10.1) software. For each emission source, all cells with assigned values ($PM_{2.5}$ emissions above zero) were plotted according to the amount of $PM_{2.5}$ generated by the source per cell.

**2.3 Validation of emission estimation factors, ground surveys, and sensitivity analysis**

Ground surveys were conducted to validate the emission estimation factors used to modify activity data factors (represented
by A in the general equation in Eq. (1)). This process was used as a form of sensitivity analysis, to determine if the reported factors originally used in the study are within ideal specifications. For reference, the sensitivity analysis procedure reported by proponents of the Clean Air for Smaller Cities project (ASEAN-GIZ) used a margin of 5% to determine variability of traffic data collection while surveying roads for mobile air emissions (Yuberk and Cornet, 2013).

A total of 98 respondents (32 for household fuels, 33 for tricycles, and 33 for PUVs) were surveyed for the validation of EEFs involving household fuels, tricycle, motorcycle, and PUV (jeeps/vans) usage. Factors included the amount and type of household cooking fuel used, registration under a tricycle/PUV riders association, trips and total travel distance per day, usage of gasoline fuels, and engine maintenance options. These factors were compared with the activity data (A) and emission reduction (ER) factors used in the general equation (Eq. (1)), and those following the standard following the
sensitivity analysis were used as EEFs used in the specific equations for household fuel usage (Eq. (2)) and tricycle fuel usage (Eq. (3)). Table 2 shows the list of EEFs that have been validated in this activity.

**Table 2.** List of emission estimation factors (EEF) validated by ground survey sensitivity analysis

| Factor | Value before validation | Value after validation | % deviation (from sensitivity analysis) |
|---|---|---|---|
| *Household fuels* | | | |
| Quantity of (household) fuel used ($Q_{fuel}$) | 194 kg yr$^{-1}$ (HECS 2011) | 173.3 kg yr$^{-1}$ | 10.7% |
| *Vehicular emissions* | | | |
| Kilometers travelled (KT) | 80 (in-house data) | 87.21 | 9.0% |
| Days in service (SDF) | 320 (in-house data) | 304.4 | 4.9% |



The group of respondents that have been surveyed for the sensitivity analysis was taken from emission "hotspots", as in, the locations where the amount of estimated PM$_{2.5}$ emissions are high. From the total possible respondents per type (household fuels, tricycles, PUVs), the sample group for this study accounts for around 1% of the total for respondents for household

fuels, around 5% for total respondents for motorcycles and tricycles, and around 2% for the total for respondents for PUVs.

## 3 Results and discussion

As seen in Fig. 5, the cells indicating the locations of household fuel-related emissions are located in the fringe of the central residential areas, where low-income households and households using charcoal as fuel are mostly situated. These cells account for possible high amounts of emissions, with levels reaching up to one kilogram of PM$_{2.5}$ per 1 hectare cell per year.

It is of note, however, that the emissions for a few of these cells is produced by commercial grilling establishments, and while these emissions were calculated differently from cells for household fuels, they are also included in the map shown in Fig. 3 due to similar emission sources and quantities.

The widespread presence of tricycles in Cabanatuan City is made evident in the map shown in Fig. 6; almost all cells aside

from those indicating open space or agricultural areas have assigned values. The emission values themselves are enough to account for a substantial fraction of the total emissions due to the high overall presence of motorcycles and tricycles as a mobile emission source in the study site.

Areas of interest concerning the very high density of tricycles and associated emissions include the central "quadrangle"

representing much of the commercial zones of the poblacion barangays of Cabanatuan City. A portion of the city center around the old capitol and the public market has a high density of tricycles contributing to PM$_{2.5}$ emissions. High concentrations of particulate emissions can also be seen in major roads extending from this central area, as well. Of notice is an isolated four-cell segment in the southwest corner of the map; this is an area near a crowded intersection of the national highway and a road leading to the central transport terminal of Cabanatuan City. In addition, this area is also a small terminal

for tricycles on its own ("*toda*") servicing the immediate vicinity and the growing commercial zone to the south of the study site.

In contrast, emissions coming from public utility vehicles (PUVs, map shown in Fig. 7) usually are found only on certain routes, as they are usually used for inter-city transport compared to tricycles. In this context, PUVs refer to vehicles often

referred to as "jeepneys" or "jeeps", but also more colloquially known as "XLTs" as they are built differently than similar vehicles used in other parts of the country such as in Metro Manila.



Emissions for PUVs are estimated to be mostly equal along major roads, such as the national road marked with cells representative of higher emissions. However, as the number of PUVs servicing the portion of the city near the study site are not as high as that of the number of tricycles, the estimated emissions generated from PUVs are relatively much lower than that of the emissions of tricycles.

A factor not usually present in major urban areas is the presence of agricultural land uses, which are more common in regional centers especially those of provincial centers. These land uses characterize cities that hybridize both rural and urban elements such as Cabanatuan City. In this context, a candidate source of $PM_{2.5}$ emissions, burning of agricultural waste, was taken into account in this spatial estimation study. Agricultural wastes such as rice straw are frequently still burned as part of

a farmland management practice in these regions, an activity that contributes to harmful emissions of particulate matter.

The map of estimated emissions from rice straw burning is shown in Fig. 8. Only a certain fraction of all agricultural land in Cabanatuan City is used in the growing of rice (this data is taken from the Cabanatuan City CLUP), and this was taken into account when estimating emissions for this map. A point to note is the fact that nearly all of the cells tagged as agricultural

are only the fringes of larger zones used for this purpose; larger agricultural areas can be found to the northwest, southeast and east of the study site. More importantly, notice that these areas are very close to the city center itself; it can be observed that the residential, commercial, and agricultural land uses are located very close to each other, almost intersecting inside the general study site.

A map showing combined emissions for all four factors used in this study is shown in Fig. 9. With the combined contributions visible in this map, areas of high concentration of $PM_{25}$ emissions become more evident. Both residential and commercial zones, as well as the dense transportation (by tricycle) network within the *poblacion* and the area immediately to its southeast contribute much of the emitted particulates; definite areas of high $PM_{2.5}$ concentrations can be seen in this location, likely from the high contributions of combustive fuels for both households and agricultural waste burning.


## 4 Summary and conclusion

As seen in the resulting maps, household fuels and tricycles account for much of the total $PM_{2.5}$ emissions in the Cabanatuan City study site. PUVs (jeeps) have a comparatively lower level of generated emissions. $PM_{2.5}$ from burning of rice straw accounts for a relatively large portion of total emissions within the study site, and is likely to account for emissions in

agricultural zones of Cabanatuan City outside the study site; this is open to future research on air quality in the city, among others.



The equation used to estimate air particulate emissions produced a value comparable to emission levels in metropolitan cities, though this is likely due to biomass-based fuels being responsible for more $PM_{2.5}$ emissions as a major source. The estimated emission levels for tricycles and rice straw burning is of note, as these two factors at their highest levels have produced emissions of at least 2 kilograms $PM_{2.5}$ per 0.01 km$^2$ (1 hectare) per year. The interface between rural and urban

land uses in a highly urbanizing city such as Cabanatuan City produces a varied environment for research on multiple fronts on possible determinants to air pollution and the monitoring of air quality in the region.

The validation of the EEFs used in the general estimation of $PM_{2.5}$ emissions have placed the actual values needed for the equations closer to the specific conditions present in Cabanatuan City; while the more generalized original in-house values

were more appropriate in areas like Metro Manila, the validation procedure has helped customize these values toward levels that are closer to what is expected in smaller cities. While the resulting modifications could only be applied to the emission factors for tricycles and household fuels, the estimation for both these sources was at least updated to more current conditions.

### 4.1 Recommendations

As stated earlier, this method for the estimation of $PM_{2.5}$ emissions is intended for use by the local governments for smaller cities and regional centers in any study country. While this method can easily be used as is for particulate matter, usage of similar methods for other components of the emission inventory process in the country (for example, usage of the method to estimate criteria pollutants) is an area of interest. As ground verification of surface features is necessary to ensure that the gridded land cover maps that will be used to determine the activity data, and as a result, the emission factors that need to be

used, the researchers recommend a detailed field survey on the ground level with surveyors equipped with GPS units, to ensure that the gathered information on surface features is up-to-date. The propagation of usage of GIS software by local government officials and non-government organizations (NGOs) specializing in air quality in small cities is another process that is both ongoing and needing more attention. This particular study has used ArcGIS, a proprietary software that requires a paid license, which may prove to be an issue for units with small financial capabilities. As this method can just as easily be

executed using free and open source GIS software such as QGIS, studies using this software may be used in the future for organizations seeking a less costly alternative for GIS. Additionally, a method for the ground verification of emission factors, similar to what was conducted here as used in the sensitivity analysis of EEFs, is highly recommended for future studies. A focus on such studies but on a much larger scale, a ground survey that represents a much larger portion of the target households/area, would be instrumental in placing the total emission estimate more accurate with regards to specific

conditions in the target city. Lastly, as this method is primarily geared towards the estimation of particulate emissions, the planning of mitigation strategies to increase air quality in target cities such as in Cabanatuan City must also be pursued in tandem with emission inventories conducted by the local government.



Acknowledgement

*This study was supported by the research grants from the Natural Sciences Research Institute (2016-ESM-001), and the Office of the Vice Chancellor for Research and Development (**141406 PNSE**), both from the University of the Philippines, Diliman. The study was also supported by the Ministry of Science, ICT and Future Planning in South Korea through the International Environmental Research Center and the UNU & GIST Joint Programme on Science and Technology for Sustainability **from 2014- 2016**. The authors would like to acknowledge the Local Government of Cabanatuan City, the Office of the City Mayor, and the Environmental Protection Division for their assistance in the activities conducted for the purposes of this study.*

Competing interests

*The authors declare that they have no conflict of interest.*

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



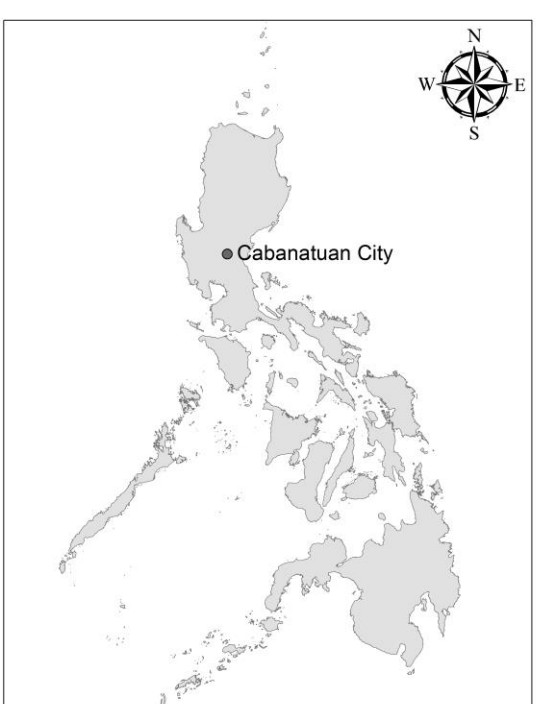

**Figure 1: Overview map of Cabanatuan City, Philippines**





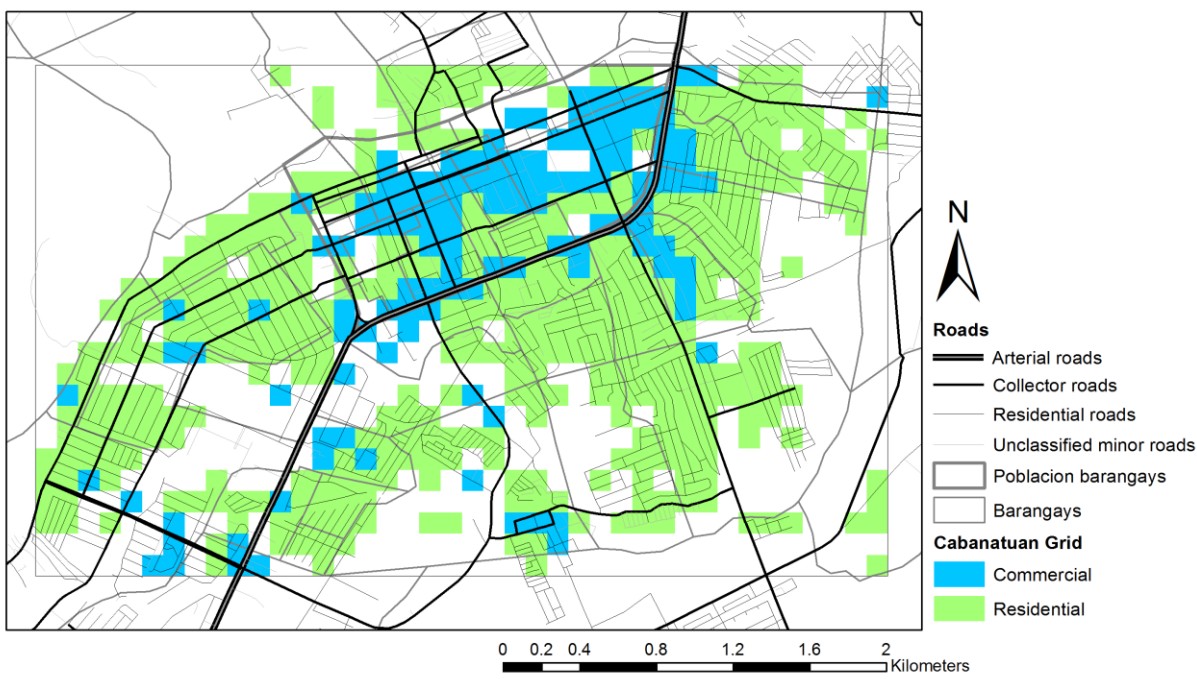

**Figure 2: The 2.4 x 4.0 km study area in Cabanatuan City containing the "city center" (*poblacion*, highlighted).**



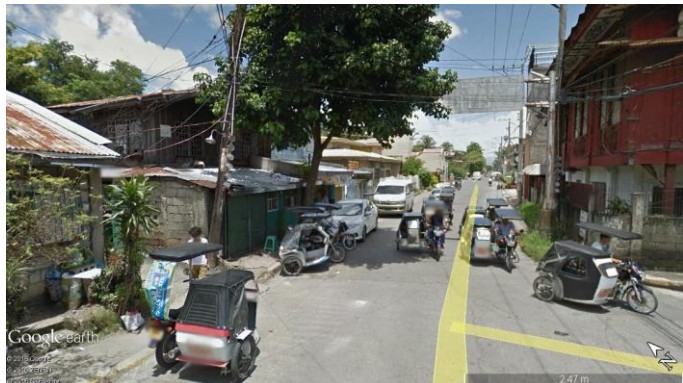

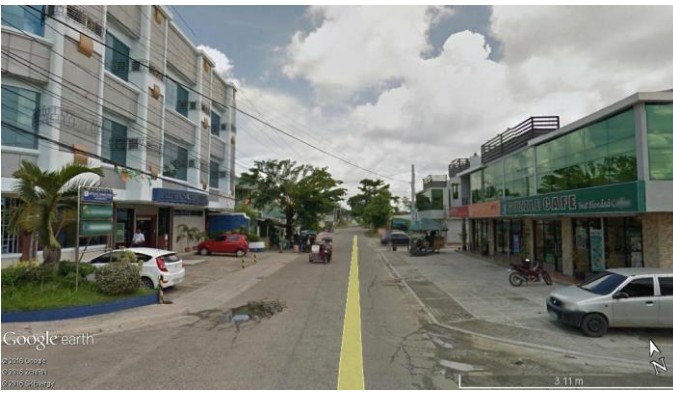

**Figure 3: Example of reference image used for Google Street View verification of surface features**





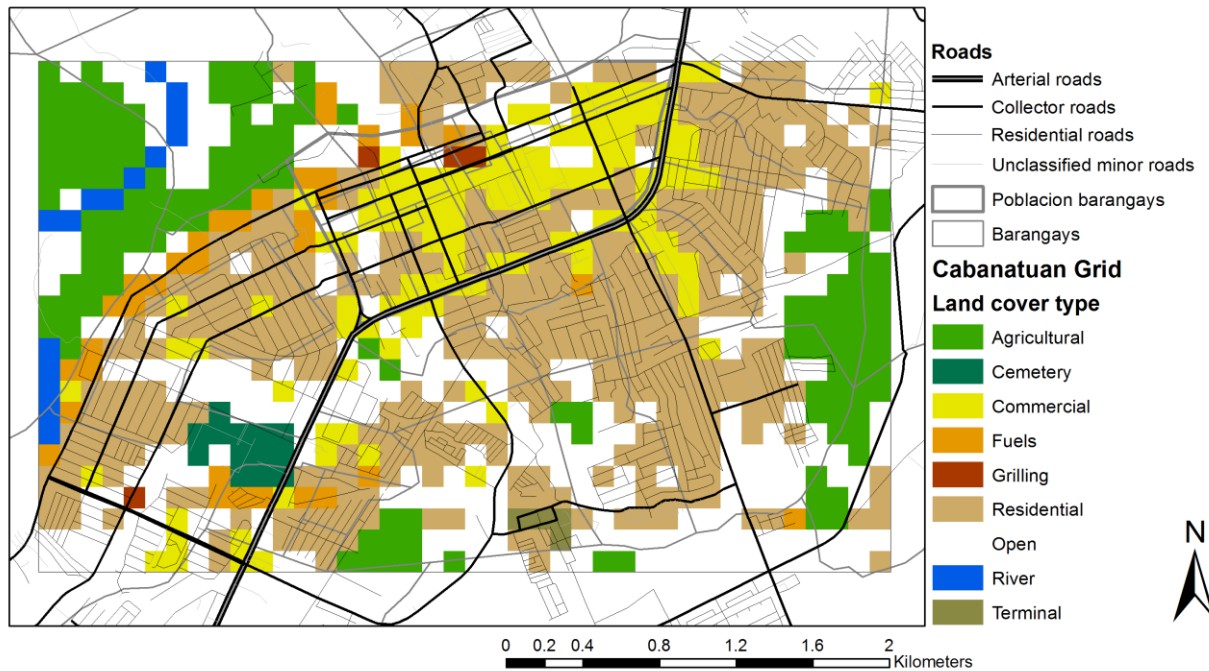

Figure 4: Land cover/land use map from interpretation of satellite image.





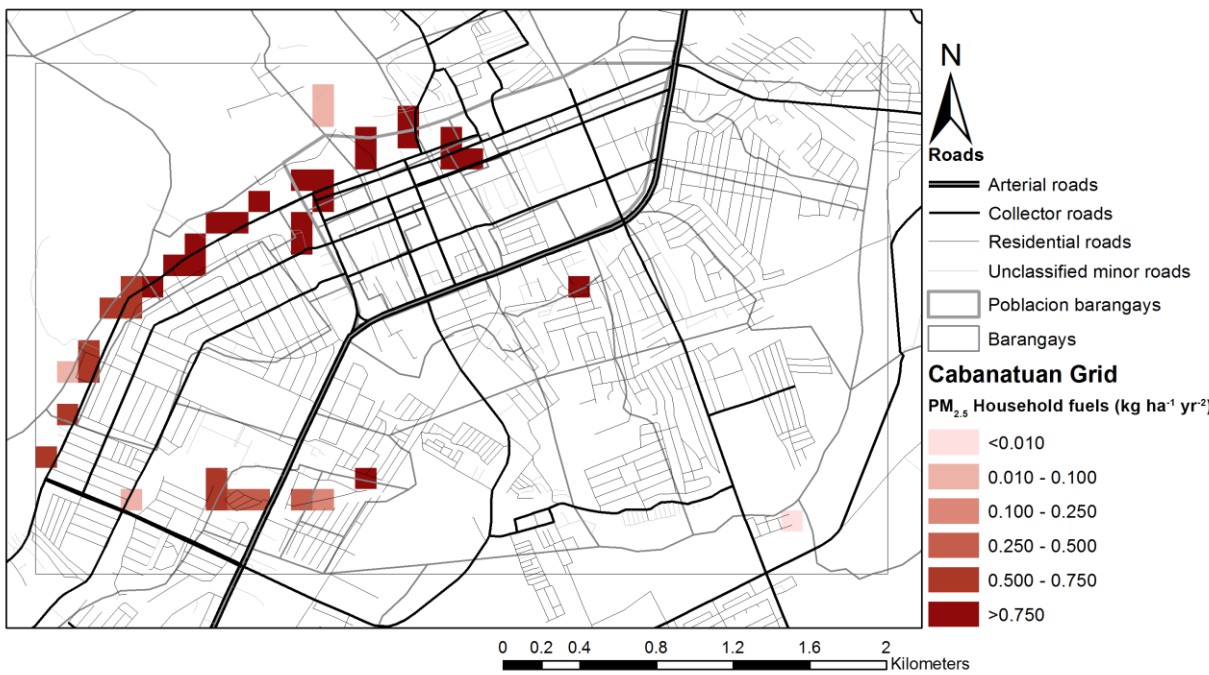

**Figure 5: Map of estimated PM$_{2.5}$ emissions from burning of household fuels.**





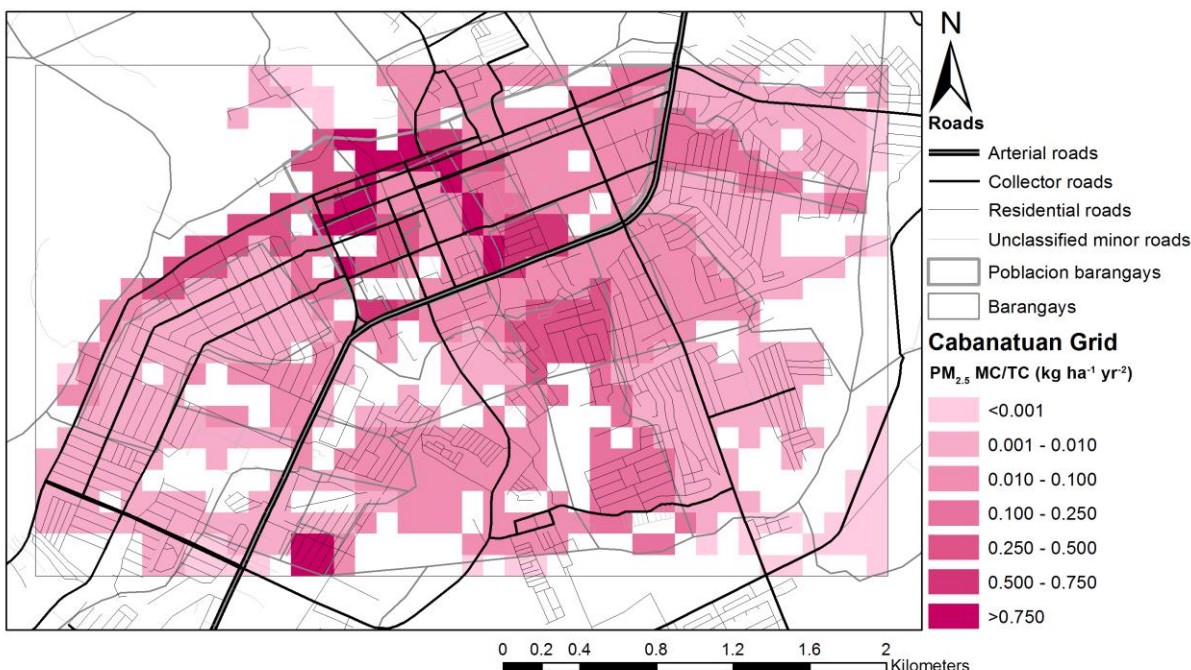

**Figure 6: Map of estimated PM$_{2.5}$ emissions from motorcycles and tricycles.**



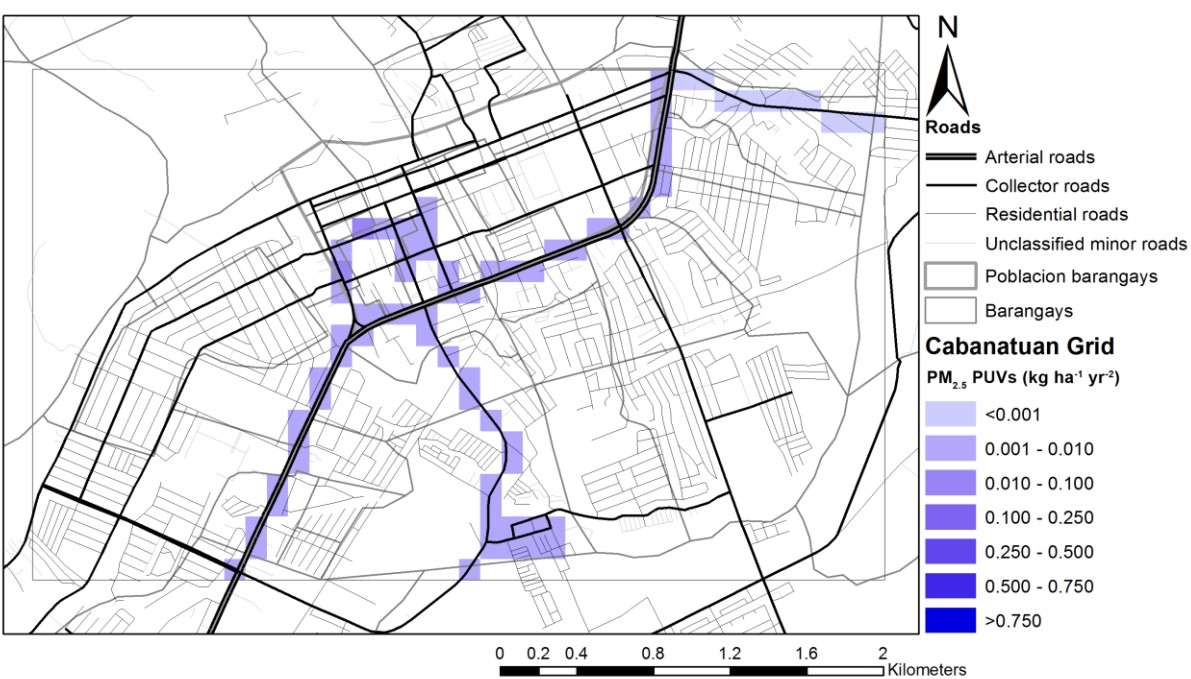

**Figure 7: Map of estimated PM$_{2.5}$ emissions from PUVs (public utility vehicles/jeepneys/XLTs).**



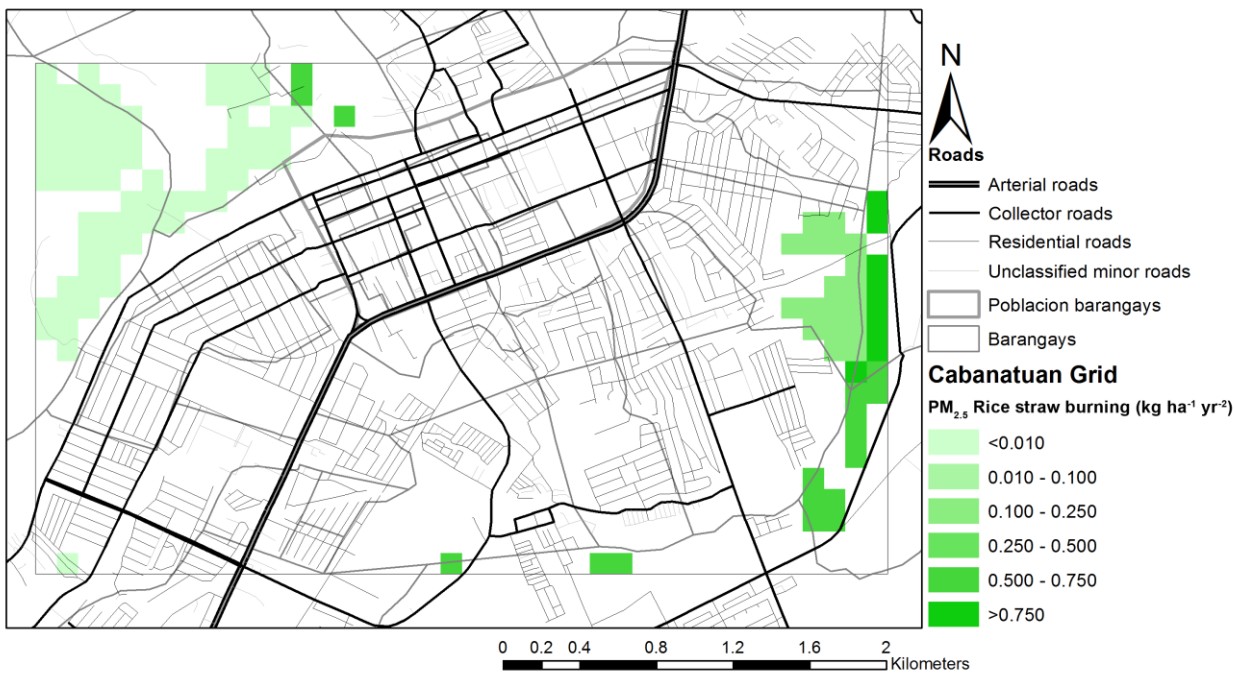

**Figure 8: Map of estimated PM$_{2.5}$ emissions from burning of rice straw as agricultural waste.**





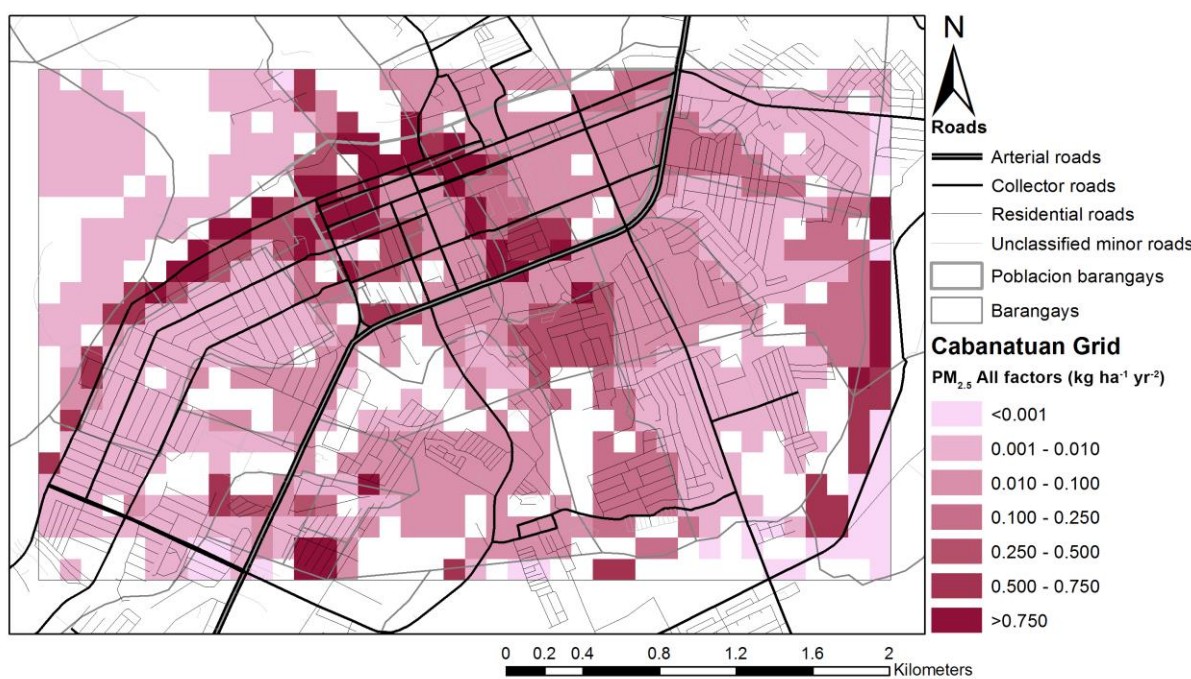

**Figure 9: Map of estimated PM$_{2.5}$ emissions combining all factors in the study.**