# Peer review of "Spatial estimation of air PM2.5 emissions using activity data, local emission factors and land cover derived from satellite imagery"

_Atmospheric Measurement Techniques, 2017_

## Referee Comment (RC1) · Anonymous Referee #1 · 19 Apr 2017

General comments

I think there are some major concerns with this manuscript that have to be taken in consideration before it can be accepted for AMT. The main problem is the language that is not clear, which means that it is difficult to fully validate the scientific content in this study. However, I think relevant scientific questions are addressed that are in the scope of AMT, but they have to better emphasise. I think the authors present a novel idea that deserve to be taken in consideration. The present method is interesting, which can also be used in the developing countries dealing with small budgets and

limitation in resources.

Major concerns

1. Due to limitation in time the review of the language has only been performed for the four first pages. Even so, it is obvious that the language has to be improved, and suggestions to improve the text are given below for these four pages. However, English is not my native language, which means that all my suggestions are probably not the best ones in an attempt to make the text more readable. The main criticism is that too much of redundant words and phrases are used in the text. However, the selection of words are also not always correct, which makes it difficult to understand the text at several places. In addition, I think the structure of the text could be improved by reducing the many paragraphs introduced. This is purely a scientific text and not a popular scientific text. At some places also very long sentences are found, which should be avoided: for example at the lines 4 – 7 on page 10. I suggest that the authors take contact with someone that is able to improve the text and/or ask AMT if they could support with this work. 2. Paragraph at lines 10 – 19: equation 3 and the corresponding text in this paragraph is very confusing. I suggest to present, where it is missing, units for the different factors included in the equations. Should the three first factors in the bracket actually be multiplied with each other? The factor SDF is not defined. Among other, the following phrase is confusing "PM2.5 per year per square kilometer per kilometer traveled". For this paragraph I will also give here an example when redundant words are used. Line start with "Emissions for motorcycles. …..", which means that you do not need to repeat this in the following sentence after the equation. The same for equations 1&4. 3. Lines 18 – 21. Concerning the low percentage values 1%, 5% and 2%, does this means that it was so few respondents that answered the survey? If so, how useful and solid is this information for the present study? You should at least make a comments on this in the manuscript.

Minor concerns

1. For E and the corresponding equations 2-4 write out the units somewhere in the text. It is not logical to name the emissions with "fuels, vehicles and straw". Maybe "households, vehicles and agricultural" instead. 2. "Figure 2. The 2.4 x 4.0 km2 study……"

Technical/language corrections

Page 1 Line 6, "Exposure to particulate matter (PM) is a serious environmental problem in many urban areas on earth." Line 8, "……involving human exposures to particulate pollutants is rare." Line 9, "fine particulate (PM2.5) emissions" Line 10, "Nueva Ecija in the Philippines," Line 11, "The emissions estimated" Line 11, "geographic information system (GIS)" Line 12, "The present results suggest that emissions from" Line 14, I think this is better "applied to any urban area, as long" Line 21, "Particulate matter, especially………haze phenomena, local and regional air quality, and climate." Line 22, "Exposure to pollutants is a risk for many people leaving in urban areas, since the level of pollution frequently exceeds WHO guidelines (Mage et al., 1996)." Line 24, "The presence of high PM2.5 is linked to increased morbidity………"

Page 2 Line 1, "carcinogenic, especially exposed for the finest fraction…." I think "for" instead of "at". Line 2. "attributed to particles acting as" Line 4 "Sources of PM2.5 are caused by many man-made activities." Line 4, "A common source of……areas is related to mobile sources, directly……" Line 7, Connect this paragraph to the previous one. Line 7, This sentence has to be improved. Line 9, Suggestion "However, PM2.5 emissions from other activities such as burning of agricultural waste occurs as well in Philippines cities………" Line 14, "At present, air quality monitoring and management are based on……" Line 15, "Standards for PM2.5 have however not been fully developed and implemented in small cities. Emissions inventories in general have likewise………in many cities." Line 17. "In addition, previous investigations are rare and limited in time, which means that temporally resolved long-term air quality monitoring data are not available." Line 20, "This study present a method to estimate PM2,5 by utilising emission factors, satellite imagery and activity data. The latter is obtained from

interpretation of geographic information system (GIS) data and by identifying and localising all sources in a city, taking into account the type of emissions (………) and activities that produces the emissions. This includes factors such as local population, density of households, number of emission-generating……" Line 27, "A limitation with this study………sources, since this is required in the mapping process." Line 30, "This study aims to determine……PM2.5, caused by individual and several aerosol sources. The present method can specifically be used for similar mixture of man-made activities as in the Philippines cities: open burning of agricultural waste and charcoal (rural activity or population) as well as usage of mobile sources (urban activity or population)."

Page 3 Line 1, Connect this paragraph with the previous one. Line 1, "Another application of this study is planning aids for local governments, as the present method can be used in emission inventories for small cities. The method was developed to be used with minimal required training and effort by stakeholders, in order to create emission inventories of aerosol sources in the cities." Line 8, "Philippines (Fig. 1)." Line 9, "and an estimated population of 296,584 in 2012." Line 10, "around half each of the total population (Cabanatuan City SEP, 2015)." Line 13, "A 2.4 by 4.0 kilometre area including the city centre and its nearest environs was selected as the study area." Line 14, "of the study area shown in Fig. 2." Line 15, As it is written, marked with grey is not shown in Fig. 2 and what is meant with "point of reference"? I have difficult to understand this sentence. Line 17, "The investigation area includes residential and commercial quarter, and even agricultural areas with less than two kilometres to a main road." Line 19, "A commercial zone and the main industrial district in Cabanatuan City located south and about 8 km from the eastern border of the investigation area, respectively, are not taken in consideration in the study." Line 22, "The investigation area was divided with 24 x 40 grid cells (100 x 100 m or 1 ha / 0.01 km2). For each cell, the type of man-made activity……Detailed images over the ground, taken by Google Street View (examples are shown in Fig. 3), were also………(residential/commercial)." Line 26. "Satellite images were dated 3 March 2016, while street view images……September 2015. Additionally, maps from OpenStreetMap were also used for identifying special landmarks or since it occasionally present more updated information than Google Street View." Line 30, "Google Earth Images have been used here instead of raw image data from example the Landsat satellite. This is because the method developed in this study is intended. . . . . .familiar with processing of satellite raw imagery data. The Google Earth images have been processed to exclude the presence of clouds and corrected for aberrations from the camera taken the satellite images." If the images really show some clouds sometimes please modify the latter sentence suggested.

Page 4 Line 2, "These images are not representative for the most current features on the ground, minor . . . . . .coordinates. It is also difficult to get access to the metadata of the original images. Even so, the Google product is useful enough and then also for the uninitiated considering the present purpose. In addition, other programs such as the Google Street View or OpenStreetMap (community-based initiative) for mapping can be used." Line 6, Sentence starting with "Actual verification. . . ." is hard to understand. Line 10, "PM2.5 emissions in the Cabanatuan city highly depend on local activity. Therefore, each grid cell (100 x 100 m) within the study area has been classified with respect to the land cover features, i.e. residential/commercial quarter, agricultural areas or other surface characteristics. Figure 4 shows that residential land use (households using liquefied petroleum gas as a fuel) are spread widely, although with noticeable commercial districts and open fields (not settled) located within this area. Two large agricultural areas are found in the northwest and east, occupied by small households likely using fuels." Improve the latter with just writing "fuels". Line 19, The Pampa River is marked with blue color in the figure, and in southeast a new residential area near open fields and agricultural areas has been built-up." Line 23, Connect this paragraph to the previous one. Line 23, "Note that some of the grid cells are marked as land uses directly: cemetery and terminal. The latter corresponding to the central transport terminal of Cabanatuan city, where high vehicular emissions are expected." Line 27, "Estimation of PM2,5 emission Line 28, "All calculations that have been used here to estimate PM2.5 emissions are based on. . . . . .(EPA, 1995): Lines 28 and 31, Emissions

of what? Line 31, "where E is equal to emissions, A is the activity rate/data (e.g. quantity of fuel, percentage of households using fuel), EF represents the. . . . . . . . ."

---

## Referee Comment (RC2) · Anonymous Referee #2 · 28 May 2017

This is an interesting paper for those researchers interested in PM2.5 emissions and learns about approaches to estimate the spatial distribution of emissions using activity data, local emission factors and land cover derived from satellite imagery. That would be of interest to the Atmospheric Measurement Techniques readership. However, the manuscript needs to be considerably improved before publication, both from the point of view of its presentation and from the amount of details provided on the data. I think the paper should be accepted after the comments and suggestions below and those

[Figure]

from the other reviewer have been addressed.

Major issues: If the paper is to be published in AMT, I advise a significant revision and restructuring of the manuscript. It was at times difficult to read. The largest issue for me is that the methods section is extremely difficult to follow. The used methods of the paper must be written clearly and explicitly. I would suggest restructuring the article to better streamline the material. There is a wide combination of methods, calculations and data products used. For example, the description of the study area and Google satellite image are first introduced in Section 2.1. And additionally, the used methods have been mentioned in the same Section 2.1. Then, all details of the activity data and emission estimations are given throughout Section 2.2. My suggestion to improve readability and clarity would be to reorganize all the methods and results into the following Sections: 2. Materials and methods 2.1 Study area 2.2 Activity data (with used data and methods) 2.3 Local emission factors (with used data and methods) 2.4 Land cover classifications by using satellite imagery (with used data and methods) 2.5 Validation of emission estimation factors, ground surveys, and sensitivity analysis 3 Results and discussion 3.1 The utilizing of activity data (with the discussions) 2.3 The utilizing of local emission factors (with the discussions) 2.4 The utilizing of Land cover classifications (with the discussions) 4 Summary and conclusion The Section "4.1 Recommendations" just stand there or there are other sessions such as 4.2, 4.3? If not, it must be done with the Section 4.

The other prominent issue I have is the not precise definition of "activity data" throughout the manuscript. In page 5 (line 5-6), the "activity data" is written as follows: "this study uses "activity data" to describe this and other relevant factors pertaining to the quantity of fuel used and percentage of households using fuel". Are the activity data estimated? And what are the significant influencing factors of the on-site specific activity data? An important concern is the emission factor. It is not clear, what is the dependence of emission factors on the fuel types. Another problem I have is that there is a little-to-no mention about the used method of land cover classification. In my opinion, the authors not clearly discussed the limitation of Google Earth. It is not clear to me whether there was used any classification method for the land cover classifications. If not, then I think a more significant treatment of the uncertainty in the classification is required. Is there the coordinate transformation considered?

Specific comments: The other reviewer provides excellent comments related to the technical correction that should be taken into account in the revision of the manuscript.

---

## Author Response (AR1)

Journal: AMTD/AMT

**Title:** Spatial estimation of air PM2.5 emissions using activity data, local emission factors and land cover derived from satellite imagery

Author(s): Hezron Gibe and Mylene Cayetano

5 **MS No.:** amt-2017-14

10

15

30

Subject: Point-by-point reply to comments (discussion papers stage)

We thank the anonymous reviewers for taking time to review this discussion paper. Since many major edits were suggested, the entire paper was edited for clearer wording and the clarification/addition of some points presented by the referees. The structure of some sections was also changed. We believe that these suggestions are important in increasing the quality of the text for recommendation for this paper to be published to AMT.

Kindly refer to the following point-by-point replies to the reviewer comments, and we appreciate your kind consideration and highly detailed comments, for improving the content and preparing this discussion paper for publication to the journal.

**I. Author's comment:**

An entry in page 1, line 4 has been corrected to show the full name of the Institute of Environmental Science and Meteorology, University of the Philippines-Diliman, where the researchers are affiliated.

**II. Evaluation and response to interactive comment by anonymous referee #1:**

**General comments**

I think there are some major concerns with this manuscript that have to be taken in consideration before it can be accepted for AMT. The main problem is the language that is not clear, which means that it is difficult to fully validate the scientific content in this study. However, I think relevant scientific questions are addressed that are in the scope of AMT, but they have to better emphasise. I think the authors present a novel idea that deserve to be taken in consideration. The present method is interesting, which can also be used in the developing countries dealing with small budgets and limitation in resources.

**25 Major concerns**

1. Due to limitation in time the review of the language has only been performed for the four first pages. Even so, it is obvious that the language has to be improved, and suggestions to improve the text are given below for these four pages. However, English is not my native language, which means that all my suggestions are probably not the best ones in an attempt to make the text more readable. The main criticism is that too much of redundant words and phrases are used in the text. However, the selection of words are also not always correct, which makes it difficult to understand the text at several places. In addition, I think the structure of the text could be improved by reducing the many paragraphs introduced. This is purely a scientific text and not a popular scientific text. At some places also very long sentences are found, which should be avoided: for example at the lines 4 - 7 on page 10. I suggest that the authors take contact with someone that is able to improve the text and/or ask AMT if they could support with this work.

- **Response:** In general, effort was taken to improve the wording of all sentences in the text. This is especially edited with the goal of reducing redundancies in some explanations found in the manuscript itself. Paragraph lengths were shortened in general, as well as splitting long sentences, found in almost all the newly edited sections of the manuscript. Specific details as to what changed can be found in later comments.
- 2. Paragraph at lines 10 19: equation 3 and the corresponding text in this paragraph is very confusing. I suggest to present,
  where it is missing, units for the different factors included in the equations. Should the three first factors in the bracket actually be multiplied with each other? The factor SDF is not defined. Among other, the following phrase is confusing "PM2.5 per year per square kilometer per kilometer traveled". For this paragraph I will also give here an example when redundant words are used. Line start with "Emissions for motorcycles. . ...", which means that you do not need to repeat this in the following sentence after the equation. The same for equations 1&4.
- 45 **Response:** The authors have reworded the section in question. Several major edits were made, the most obvious one the splitting of the former equation (3) to equations (3) and (4). Wording was changed to reflect a focus on "vehicular sources" of PM2.5. Most of the ambiguous factors in question were those intended to serve as the activity data factors for tricycles.

NAF in the previous version was renamed to AVF (association vehicles factor) for clarity. Units were added to the explanation of all emission factor estimation equations (1-5). The new explanation hopefully makes it clear as to why the first three factors ( $N_u$ , DF, AVF) should be multiplied. The definition for factor SDF (distance/kilometers traveled) was also added. Similar edits were also used for sections containing equations (1) and (4) (now (1) and (5))

**Page 6, Lines 11-20:**  $PM_{2.5}$  emissions for vehicular sources were estimated with the formula shown in Eq. (3) and Eq. (4).

$$E_{MC/TC} = (N_u \times DF \times AVF) \times (EF \times KT \times SDF) \times 0.01,$$

$$E_{PUV} = (N_u \times DF) \times EF \times 0.01,$$
(3)
(4)

Factors that are the same for both equations include:  $N_u$ , the estimated number of vehicle units, DF, the density factor (amount of vehicles per km2), and EF, the emission factor. The in-house emission factor for MC/TCs is measured as PM2.5 per kilometer traveled (per vehicle). Due to this non-standard EF unit, additional factors are required in Eq. (3). These include the association vehicles factor (AVF), the percentage of vehicles which are officially registered and properly accounted for by the city. To scale the EF to its proper units, it is multiplied by factor KT (kilometers traveled per day) and SDF (days in service per year). Similar to the previous example, the total is also multiplied by 0.01 to scale to each 0.01 km2 cell. The DF and NAF was verified using sensitivity analysis by ground surveys as detailed in section 2.4.

**Page 4, Lines 17-26:** All calculations that have been used to estimate  $PM_{2.5}$  emissions are based on a general formula used by the US EPA in the AP 42 Compilation of Air Pollutant Emission Factors (EPA, 1995), as shown in Eq. (1)

$$E = A \times EF \times \left(1 - \frac{ER}{100}\right),\tag{1}$$

where: E is equal to  $PM_{2.5}$  emissions, A is the activity rate/data (e.g. quantity of fuel used, percentage of households using fuel), EF represents the emission factor, and ER is the overall emission reduction factor/efficiency in percent, if applicable. In the present method, E is estimated as being the quantity of  $PM_{2.5}$  per unit cell: micrograms per 0.01 km2 (1 hectare) per year. ER refers to other factors affecting the total amount of  $PM_{2.5}$  emissions (such as factors not directly accounting towards the quantity of fuel used; ER factors also incorporate the activity of those using quantities of fuel lower than average). This comprises the various factors that are also part of activity data (as in, factors that modify the amount of emissions generated) as used in this study.

Page 6, Lines 20-27: Emissions for agricultural waste burning were estimated with the formula shown in Eq. (5):

$$E_{agricultural} = {\binom{RS}{RA}} \times EF \times SF , \qquad (5)$$

where: RS is the amount of rice straw produced per year, divided by RA, which is the total area in hectares (0.01 km2) used for growing of rice. EF is the in-house obtained emission factor for rice straw burning  $PM_{2.5}$  per year per square kilometer. SF is the survey factor, representing the percentage of farming area where burning of rice straw as agricultural waste is used. This reduction factor is taken from the study of Launio, et al. (2013).

3. Lines 18 - 21. Concerning the low percentage values 1%, 5% and 2%, does this means that it was so few respondents that answered the survey? If so, how useful and solid is this information for the present study? You should at least make a comments on this in the manuscript.

35 **Response:** Edited paragraph starting in page 7, line 19 to comment on this. Also, an edit was made to the paragraph starting in page 10, line 4 as an additional comment:

**Page 7, Line 19 – Page 8, Line 2:** The respondents that were surveyed were taken from specific areas, termed emission hotspots. These are locations where the amount of estimated  $PM_{2.5}$  emissions are expected to be high. From the total estimated maximum respondents per type (households, vehicles (MC/TCs, PUVs)), the sample group for this study accounts for around 1% of the total for respondents for households, around 5% for total respondents for MC/TCs, and around 2% for the total for respondents for PUVs. This proportion of the sample size is very low, so the proponents have implemented stratified sampling intended to make the small sample as representative of the entire study area as possible.

**Page 10, Lines 4-10:** The validation of specific activity data factors is effective at adapting them closer to the specific conditions present in Cabanatuan City. While the more general original in-house values are more appropriate in areas like Metro Manila, the validation procedure has made them more appropriate for smaller cities in general. An issue during the ground survey activity involves its small sample size compared to the possible maximum number of respondents in the

5

15

10

20

25

30

40

investigation area. However, the benefits of fine-tuning the activity data with this analysis outweigh its disadvantages. Also, in future researches, the ground survey and sensitivity analysis validation will highly be improved if the sample size is greatly increased.

**5 Minor concerns**

15

20

25

1. For E and the corresponding equations 2-4 write out the units somewhere in the text. It is not logical to name the emissions with "fuels, vehicles and straw". Maybe "households, vehicles and agricultural" instead.

**Response:** Relevant sections were edited to include units for all factors. The names of the E factors (i.e.  $E_{households}$ ) for all equations were also changed to reflect this.

(2)

10 Page 6, Lines 2-27: Emissions for household fuel (charcoal) were estimated with the formula shown in Eq. (2):

 $E_{households} = (N_h \times HF) \times Q_{fuel} \times EF \times 0.01$ ,

where:  $N_h$  is the estimated number of households (generated from city government data), and HF is the percentage of all households using charcoal as fuel, obtained from the HECS.  $Q_{fuel}$  is the quantity of fuel in kilograms used per year by each household, sourced from the HECS and verified using sensitivity analysis by ground surveys (see section 2.4). EF corresponds to the emission factor for charcoal fuel  $PM_{2.5}$  per square kilometer per year; this is then multiplied by 0.01 to scale to each 0.01 km2 cell.

PM2.5 emissions for vehicular sources were estimated with the formula shown in Eq. (3) and Eq. (4).

$$E_{MC/TC} = (N_u \times DF \times AVF) \times (EF \times KT \times SDF) \times 0.01,$$

$$E_{PUV} = (N_u \times DF) \times EF \times 0.01,$$
(3)
(4)

Factors that are the same for both equations include:  $N_u$ , the estimated number of vehicle units, DF, the density factor (amount of vehicles per km2), and EF, the emission factor. The in-house emission factor for MC/TCs is measured as  $PM_{2.5}$ per kilometer traveled (per vehicle). Due to this non-standard EF unit, additional factors are required in Eq. (3). These include the association vehicles factor (AVF), the percentage of vehicles which are officially registered and properly accounted for by the city. To scale the EF to its proper units, it is multiplied by factor KT (kilometers traveled per day) and SDF (days in service per year). Similar to the previous example, the total is also multiplied by 0.01 to scale to each 0.01 km2 cell. The DF and NAF was verified using sensitivity analysis by ground surveys as detailed in section 2.4.

Emissions for agricultural waste burning were estimated with the formula shown in Eq. (5):

$$B0 E_{agricultural} = \left(\frac{RS}{RA}\right) \times EF \times SF , (5)$$

where: RS is the amount of rice straw produced per year, divided by RA, which is the total area in hectares (0.01 km2) used for growing of rice. EF is the in-house obtained emission factor for rice straw burning  $PM_{2.5}$  per year per square kilometer. SF is the survey factor, representing the percentage of farming area where burning of rice straw as agricultural waste is used. This reduction factor is taken from the study of Launio, et al. (2013).

35 **Authors' comment:** The following corrections suggested by anonymous referee #1 were made in various capacities, taking into account our intent for the study methods, and acknowledging our own writing style and use of the English language.

Corrections suggested by anonymous referee #1: 2. "Figure 2. The 2.4 x 4.0 km2 study. . . . . "

Response: Caption edited for technical purposes

40 **Page 14, Line 2 (caption):** Figure 2: The 2.4 x 4.0 km study area in Cabanatuan City containing the "city center" (poblacion, highlighted).

Technical/language corrections Page 1 Line 6, "Exposure to particulate matter (PM) is a serious environmental problem in many urban areas on earth." Line 8, ". . . . . involving human exposures to particulate pollutants is rare." Line 9, "fine

particulate (PM2.5) emissions" Line 10, "Nueva Ecija in the Philippines," Line 11, "The emissions estimated" Line 11, "geographic information system (GIS)" Line 12, "The present results suggest that emissions from" Line 14, I think this is better "applied to any urban area, as long"

Response: Abstract section mostly edited as suggested, see full changes below:

- 5 Page 1, Lines 6-15: Exposure to air particulate matter (APM) is a serious environmental problem in many urban areas on Earth. In the Philippines, most existing studies and emission inventories have mainly focused on point and mobile sources, while research involving human exposures to particulate pollutants is rare. This paper presents a method for estimating the amount fine particulate (PM2.5) emissions in a test study site in Cabanatuan City, Nueva Ecija in the Philippines, by utilizing local emission factors, regionally procured data and land cover/land use (activity data) interpreted from satellite imagery. Geographic information system (GIS) software was used to map the estimated emissions in the study area. The present results suggest that vehicular emissions from motorcycles and tricycles, as well as fuels used by households (charcoal) and burning of agricultural waste largely contribute to PM2.5 emissions in Cabanatuan City. Overall, the method used in this study can be applied in other small urbanizing cities, as long as on-site specific activity data, emission factor and satellite-imaged land cover are available.
- Line 21, "Particulate matter, especially. . . . . ...haze phenomena, local and regional air quality, and climate." Line 22, "Exposure to pollutants is a risk for many people leaving in urban areas, since the level of pollution frequently exceeds WHO guidelines (Mage et al., 1996)." Line 24, "The presence of high PM2.5 is linked to increased morbidity. . . . . . ."

Response: Introduction section (paragraph beginning in page 1, line 21) was edited as suggested.

30

Page 1, Line 21 – Page 2, Line 2: Exposure to air particulate matter, especially fine particles smaller than 2.5 micrometers
in size (PM2.5), can reduce air quality, affect visibility through smog and other haze phenomena, and introduce lasting effects on climate on a local and regional scale. Exposure to pollutants is a risk for many people living in urban areas, since the level of pollution frequently exceeds WHO guideline values (Mage, et al., 1996). The presence of PM2.5 is linked to increased morbidity and mortality risk, especially in incidences of various cardio-pulmonary diseases (Chen, et al., 2008; Lin, et al., 2016; Wu, et al., 2013), birth defects (Goto, et al., 2016), and cancer (Cassidy, et al., 2007). PM2.5 pollution is also considered carcinogenic, especially exposure to the finest fractions (ultrafine particles) (Bocchi, et al., 2016). This can be attributed to particles acting as carriers of mutagenic and genotoxic compounds (Chen, et al., 2016).

Page 2 Line 1, "carcinogenic, especially exposed for the finest fraction. . .." I think "for" instead of "at". Line 2. "attributed to particles acting as" Line 4 "Sources of PM2.5 are caused by many man-made activities." Line 4, "A common source of. . . . . . areas is related to mobile sources, directly. . . . ." Line 7, Connect this paragraph to the previous one. Line 7, This sentence has to be improved. Line 9, Suggestion "However, PM2.5 emissions from other activities such as burning of agricultural waste occurs as well in Philippines cities. . . . .."

**Response:** Various edits for wording, clarity, and content were made to the paragraph beginning in page 2, line 4 as suggested (some edits are not exactly the same as suggested by anonymous referee #1)

Page 2, Lines 4-9: Sources of PM2.5 are caused by many man-made activities. A common source of PM2.5, in urban areas is related to mobile sources, directly emitted by internal combustion processes inside vehicles of all types (Andrade, et al., 2012; Ahanchian and Biona, 2014; Chen, et al., 2016). In most of the reports from Philippine cities, vehicular emissions reported in inventories use foreign emission factors (such as CORINAIR and AP 42). However, PM2.5 emissions from other activities such as burning of agricultural waste occurs as well in cities with a mixture of rural and urban land uses (Sarigiannis, et al., 2014; Kim Oanh, et al., 2011; Gadde, et al., 2009).

40 Line 14, "At present, air quality monitoring and management are based on. . . . . ." Line 15, "Standards for PM2.5 have however not been fully developed and implemented in small cities. Emissions inventories in general have likewise. . . . . . in many cities." Line 17. "In addition, previous investigations are rare and limited in time, which means that temporally resolved long-term air quality monitoring data are not available."

**Response:** Various edits for wording, clarity, and content were made to the paragraph beginning in page 2, line 11 as suggested.

**Page 2, Lines 11-14:** At present, air quality monitoring and management are based on  $PM_{10}$  and total suspended particles (*TSP*) as an indicator. Standards for  $PM_{2.5}$  have however not been fully developed and implemented in small cities. Emission inventories in general have likewise not been conducted in many cities. In addition, previous investigations are rare and limited in time, which means that temporally resolved long-term air quality monitoring data are not available.

Line 20, "This study present a method to estimate PM2,5 by utilising emission factors, satellite imagery and activity data. The latter is obtained from interpretation of geographic information system (GIS) data and by identifying and localising all sources in a city, taking into account the type of emissions  $(\ldots \ldots \ldots)$  and activities that produces the emissions. This includes factors such as local population, density of households, number of emission-generating. ....." Line 27, "A limitation with this study......sources, since this is required in the mapping process."

**Response:** Various edits for wording, clarity, and content were made to the paragraph beginning in page 2, line 16 as suggested (spelling differences reflect local usage of English).

Page 2, Lines 16-22: This study presents a method to estimate PM2.5 by utilizing locally sourced emission factors, satellite imagery, and activity data. The latter is obtained from interpretation of geographic information system (GIS) data and by identifying and localizing all sources in the city, taking into account the type of emission (point, area, mobile), and activities which produces the emissions. This includes factors such as local population, density of households, number of emission generating events, and the type and amount of various fuels used. This, in conjunction with various local emission factors, will be used to estimate total PM2.5 emissions. A limitation of this study is that all emission sources are treated as being area sources, since this is required in the mapping process.

5

Line 30, "This study aims to determine. . ... ...PM2.5, caused by individual and several aerosol sources. The present method can specifically be used for similar mixture of man-made activities as in the Philippines cities: open burning of agricultural waste and charcoal (rural activity or population) as well as usage of mobile sources (urban activity or population)." Page 3 Line 1, Connect this paragraph with the previous one. Line 1, "Another application of this study is planning aids for local governments, as the present method can be used in emission inventories for small cities. The method was developed to be used with minimal required training and effort by stakeholders, in order to create emission inventories of aerosol sources in the cities."

**Response:** Various edits for wording, clarity, and content were made to the paragraph beginning in page 2, line 24 as suggested (some edits are not exactly the same as suggested by anonymous referee #1)

Page 2, Lines 24-32: From the resulting maps, the study aims to determine areas of high concentration of PM2.5, caused by individual and several aerosol sources. The present method can specifically be used for similar mixtures of man-made activities present in Philippine cities. This method is specifically meant to explore this method for use in relatively small regional urban centers and cities in the Philippines; especially due to these cities being situated in locations where there is a mixture of rural and urban activities. Sources corresponding to rural activity include open burning of agricultural waste and the usage of household cooking fuels such as charcoal. Sources corresponding to urban activity include vehicular mobile sources such as tricycles, jeepneys, and PUVs (buses and vans). Another application for this study is planning aids for local governments; as the present method can be used in emission inventories for small cities. The method was developed to be used with minimal required training and effort by stakeholders, in order to create emission inventories of aerosol sources in the cities.

Line 8, "Philippines (Fig. 1)." Line 9, "and an estimated population of 296,584 in 2012." Line 10, "around half each of the total population (Cabanatuan City SEP, 2015)." Line 13, "A 2.4 by 4.0 kilometre area including the city centre and its nearest environs was selected as the study area." Line 14, "of the study area shown in Fig. 2." Line 15, As it is written, marked with grey is not shown in Fig. 2 and what is meant with "point of reference"? I have difficult to understand this sentence. Line 17, "The investigation area includes residential and commercial quarter, and even agricultural areas with less than two kilometres to a main road." Line 19, "A commercial zone and the main industrial district in Cabanatuan City located south and about 8 km from the eastern border of the investigation area, respectively, are not taken in consideration in the study."

**Response:** Various edits for wording, clarity, and content were made to the paragraph beginning in page 3, line 3 as suggested (some edits are not exactly the same as suggested by anonymous referee #1, some spelling differences reflecting local usage of English)

45 **Page 3, Lines 3-14:** The test study was conducted in Cabanatuan City, Philippines (Fig. 1). It is the former capital and largest city of the province of Nueva Ecija, with a land area of 190.67 square kilometers and an estimated population of 296,584 in 2012. On average, the population density is around 1,516 persons per square kilometer. The urban and rural population take up around half each of the total population (Cabanatuan City SEP, 2015).

A 2.4 by 4.0 kilometer area including the city center and its nearest environs was selected as the main study area. The town proper, (locally known as the poblacion) is highlighted in the map of the study area shown in Fig. 2. Grey lines indicate

boundaries of barangays (the smallest administrative division of a local government, a similar concept to town wards or districts), and the constituent barangays of the poblacion are marked using thicker grey outlines. The investigation area includes residential and commercial zones, and even agricultural areas less than two kilometers away from a main road. A commercial zone and the planned main industrial district in Cabanatuan City located south and about 8-10 km southeast of the investigation area, respectively, are not taken into consideration in the study.

5

10

15

20

25

30

50

**Response:** Various edits for wording, clarity, and content were made to the paragraph beginning in page 3, line 16 as suggested. Some technical edits are also present (some edits are not exactly the same as suggested by anonymous referee #1)

**Page 3, Lines 16-24:** The investigation area was divided with 24 x 40 grid cells (100 x 100 m or 1 ha / 0.01 km2 each). For each cell, the type of man-made activity was interpreted from satellite images taken from Google Earth software. The classification process is similar to what is done for methods of supervised classification of land cover. The image of the surface feature is compared to a reference area of known land cover. Due to the size of each cell, the detail of each ground feature can be clearly seen. Detailed images over the ground, taken by Google Street View (examples are shown in Fig. 3) was used to verify building types (residential/commercial). Satellite images were dated 3 March, 2016, while ground level (Street View) images were dated September 2015. Additionally, maps from OpenStreetMap were also used for identifying special landmarks or as an additional resource since it occasionally presents more updated information on surface features than Google Street View.

Line 30, "Google Earth Images have been used here instead of raw image data from example the Landsat satellite. This is because the method developed in this study is intended. . . . . .familiar with processing of satellite raw imagery data. The Google Earth images have been processed to exclude the presence of clouds and corrected for aberrations from the camera taken the satellite images." If the images really show some clouds sometimes please modify the latter sentence suggested.

Page 4 Line 2, "These images are not representative for the most current features on the ground, minor . . . . . . . coordinates. It is also difficult to get access to the metadata of the original images. Even so, the Google product is useful enough and then also for the uninitiated considering the present purpose. In addition, other programs such as the Google Street View or OpenStreetMap (community-based initiative) for mapping can be used." Line 6, Sentence starting with "Actual verification. . ..." is hard to understand.

**Response:** Various edits for wording, clarity, and content were made to the paragraph beginning in page 3, line 26 as suggested (some edits are not exactly the same as suggested by anonymous referee #1)

Page 3, Line 26 – Page 4, Line 6: Google Earth images have been used here instead of raw image data from, for example, the Landsat satellite (The collaged image used in Google Earth is sourced from processed images from Landsat and the 35 European Space Agency (ESA)'s Copernicus program). This is because the method developed in this study is intended to be used by personnel not necessarily familiar with processing of satellite raw imagery data. The Google Earth images have been processed to minimize the presence of clouds and corrected for aberrations from the camera taking the satellite images. These images are not representative of the most current features on the ground. There is also a slight deviation of the actual coordinates representing the location of the area due to the orthographic projection of the satellite image. This is consistent 40 with geolocation deviations present in most consumer-grade satellite/GPS products. It is also difficult to get access to the metadata of the original images. Even so, the Google satellite image product is useful enough for the uninitiated considering the present purpose. In addition, other data products such as Google Street View or OpenStreetMap (community-based initiative) can be used. The usage of supporting documents such as existing local government land use plans and land cover maps, as well as actual verification of features at the ground level (ground truth, that is, information on surface features in 45 the study area), is necessary, and was used in this study to verify land cover and land use features at the surface level.

Line 10, "PM2.5 emissions in the Cabanatuan city highly depend on local activity. Therefore, each grid cell (100 x 100 m) within the study area has been classified with respect to the land cover features, i.e. residential/commercial quarter, agricultural areas or other surface characteristics. Figure 4 shows that residential land use (households using liquefied petroleum gas as a fuel) are spread widely, although with noticeable commercial districts and open fields (not settled) located within this area. Two large agricultural areas are found in the northwest and east, occupied by small households likely using fuels." Improve the latter with just writing "fuels". Line 19, The Pampa River is marked with blue color in the

figure, and in southeast a new residential area near open fields and agricultural areas has been built-up." Line 23, Connect this paragraph to the previous one. Line 23, "Note that some of the grid cells are marked as land uses directly: cemetery and terminal. The latter corresponding to the central transport terminal of Cabanatuan city, where high vehicular emissions are expected."

- 5 **Response:** Various edits for wording, clarity, and content were made to the paragraph beginning in page 4, line 8 as suggested (some edits are not exactly the same as suggested by anonymous referee #1). The usage of the wording "household fuels" was fixed overall in this section and in some other parts of the paper to now read "households" or "fuels" depending on context instead.
- Page 4, Lines 8-15: PM2.5 emissions in Cabanatuan City highly depend on local activity. Therefore, each grid cell (100 x 100 m) within the study area has been classified with respect to the land cover features, i.e. residential/commercial zones, agricultural areas, or other surface characteristics. Figure 4 shows that residential land use (households using liquefied petroleum gas as a fuel) are spread widely, although with noticeable commercial districts and open fields (not settled or occupied) located within this area. Two large agricultural areas are found in the northwest and east, occupied by small households likely using fuels. The Pampanga River is marked in blue in the figure, and in the southeast, a new residential area near open fields and agricultural areas has been built-up. Note that some of the grid cells are marked as land uses directly: cemetery and terminal, the latter corresponding to the central transport terminal of Cabanatuan City, where high vehicular emissions are expected.

Line 27, "Estimation of PM2,5 emission Line 28, "All calculations that have been used here to estimate PM2.5 emissions are based on. . . . . (EPA, 1995): Lines 28 and 31, Emissions of what? Line 31, "where E is equal to emissions, A is the activity rate/data (e.g. quantity of fuel, percentage of households using fuel), EF represents the. . . . . . ."

**Response:** Various edits for wording, clarity, and content were made to the section beginning in page 4, line 17 as suggested (some edits are not exactly the same as suggested by anonymous referee #1)

**Page 4, Lines 17-26:** All calculations that have been used to estimate  $PM_{2.5}$  emissions are based on a general formula used by the US EPA in the AP 42 Compilation of Air Pollutant Emission Factors (EPA, 1995), as shown in Eq. (1)

25
$$E = A \times EF \times \left(1 - \frac{ER}{100}\right), \tag{1}$$

where: E is equal to  $PM_{2.5}$  emissions, A is the activity rate/data (e.g. quantity of fuel used, percentage of households using fuel), EF represents the emission factor, and ER is the overall emission reduction factor/efficiency in percent, if applicable. In the present method, E is estimated as being the quantity of  $PM_{2.5}$  per unit cell: micrograms per 0.01 km2 (1 hectare) per year. ER refers to other factors affecting the total amount of  $PM_{2.5}$  emissions (such as factors not directly accounting towards the quantity of fuel used; ER factors also incorporate the activity of those using quantities of fuel lower than average). This comprises the various factors that are also part of activity data (as in, factors that modify the amount of emissions generated) as used in this study.

**III. Evaluation and response to interactive comment by anonymous referee #2:**

This is an interesting paper for those researchers interested in PM2.5 emissions and learns about approaches to estimate the spatial distribution of emissions using activity data, local emission factors and land cover derived from satellite imagery. That would be of interest to the Atmospheric Measurement Techniques readership. However, the manuscript needs to be considerably improved before publication, both from the point of view of its presentation and from the amount of details provided on the data. I think the paper should be accepted after the comments and suggestions below and those from the other reviewer have been addressed.

**40 Major issues**

If the paper is to be published in AMT, I advise a significant revision and restructuring of the manuscript. It was at times difficult to read. The largest issue for me is that the methods section is extremely difficult to follow. The used methods of the paper must be written clearly and explicitly. I would suggest restructuring the article to better streamline the material. There is a wide combination of methods, calculations and data products used. For example, the description of the study area and Google satellite image are first introduced in Section 2.1. And additionally, the used methods have been mentioned in the same Section 2.1. Then, all details of the activity data and emission estimations are given throughout Section 2.2. My suggestion to improve readability and clarity would be to reorganize all the methods and results into the following Sections: 2. Materials and methods 2.1 Study area 2.2 Activity data (with used data and methods) 2.3 Local emission factors (with

30

45

used data and methods) 2.4 Land cover classifications by using satellite imagery (with used data and methods) 2.5 Validation of emission estimation factors, ground surveys, and sensitivity analysis 3 Results and discussion 3.1 The utilizing of activity data (with the discussions) 2.3 The utilizing of local emission factors (with the discussions) 2.4 The utilizing of Land cover classifications (with the discussions) 4 Summary and conclusion The Section "4.1 Recommendations" just stand there or there are other sessions such as 4.2, 4.3? If not, it must be done with the Section 4.

**Response:** The entire manuscript from section 2 onwards has been restructured using the following headers:

**2 Materials and methods**

- 3.1 Study area
- 3.2 Land cover classification using satellite imagery
- 3.3 PM2.5 emission estimation
  - 3.3.1 Local emission factors
  - 3.3.2 Activity data
  - **3.3.3** Emission estimation equations
  - 3.4 Validation of activity data factors (ground surveys and sensitivity analysis)

**15 **3 Results and discussion**

5

10

35

**4 Summary and conclusion**

**5** Recommendations**

**Response (continued):** This was done to help streamline section 2 in particular. New sections were added to sections 2.2, 2.3/2.3.1/2.3.2/2.3.3, and 2.4 to give more detail as to the methods used in the study.

The other prominent issue I have is the not precise definition of "activity data" throughout the manuscript. In page 5 (line 5-6), the "activity data" is written as follows: "this study uses "activity data" to describe this and other relevant factors pertaining to the quantity of fuel used and percentage of households using fuel". Are the activity data estimated? And what are the significant influencing factors of the on-site specific activity data? An important concern is the emission factor. It is not clear, what is the dependence of emission factors on the fuel types. Another problem I have is that there is a little-to-no mention about the used method of land cover classification.

**Response:** The definition of "activity data" is now worded to follow more closely with how it is used in the general EPA equation as explained in the section starting in page 4, line 17, and used as the basis for equation (1). All mentions of "emission estimation factors" or "EEF" used in the previous iteration of the manuscript were removed in favor of wording that includes the factors that make up ER in equation (1) under the definition of "activity data" as well.

30 **Page 4, Lines 17-26:** All calculations that have been used to estimate PM2.5 emissions are based on a general formula used by the US EPA in the AP 42 Compilation of Air Pollutant Emission Factors (EPA, 1995), as shown in Eq. (1)

$$E = A \times EF \times \left(1 - \frac{ER}{100}\right),\tag{1}$$

where: E is equal to  $PM_{2.5}$  emissions, A is the activity rate/data (e.g. quantity of fuel used, percentage of households using fuel), EF represents the emission factor, and ER is the overall emission reduction factor/efficiency in percent, if applicable. In the present method, E is estimated as being the quantity of  $PM_{2.5}$  per unit cell: micrograms per 0.01 km2 (1 hectare) per year. ER refers to other factors affecting the total amount of  $PM_{2.5}$  emissions (such as factors not directly accounting towards the quantity of fuel used; ER factors also incorporate the activity of those using quantities of fuel lower than average). This comprises the various factors that are also part of activity data (as in, factors that modify the amount of emissions generated) as used in this study.

In my opinion, the authors not clearly discussed the limitation of Google Earth. It is not clear to me whether there was used any classification method for the land cover classifications. If not, then I think a more significant treatment of the uncertainty in the classification is required. Is there the coordinate transformation considered?

**Response:** The new section 2.2 was created, structured, and edited to address this issue. An additional few sentences were added to the paragraph starting in page 3, line 16 to address the method used in the land cover classification.

45 **Page 3, Lines 16-24:** The investigation area was divided with 24 x 40 grid cells (100 x 100 m or 1 ha / 0.01 km2 each). For each cell, the type of man-made activity was interpreted from satellite images taken from Google Earth software. The

classification process is similar to what is done for methods of supervised classification of land cover. The image of the surface feature is compared to a reference area of known land cover. Due to the size of each cell, the detail of each ground feature can be clearly seen. Detailed images over the ground, taken by Google Street View (examples are shown in Fig. 3) was used to verify building types (residential/commercial). Satellite images were dated 3 March, 2016, while ground level (Street View) images were dated September 2015. Additionally, maps from OpenStreetMap were also used for identifying special landmarks or as an additional resource since it occasionally presents more updated information on surface features than Google Street View.

**Response** (continued): Issues regarding the usage of Google Earth images were laid out in the paragraph starting in page 3, line 26.

10 Page 3, Line 26 – Page 4, Line 6: Google Earth images have been used here instead of raw image data from, for example, the Landsat satellite (The collaged image used in Google Earth is sourced from processed images from Landsat and the European Space Agency (ESA)'s Copernicus program). This is because the method developed in this study is intended to be used by personnel not necessarily familiar with processing of satellite raw imagery data. The Google Earth images have been processed to minimize the presence of clouds and corrected for aberrations from the camera taking the satellite images. 15 These images are not representative of the most current features on the ground. There is also a slight deviation of the actual coordinates representing the location of the area due to the orthographic projection of the satellite image. This is consistent with geolocation deviations present in most consumer-grade satellite/GPS products. It is also difficult to get access to the metadata of the original images. Even so, the Google satellite image product is useful enough for the uninitiated considering the present purpose. In addition, other data products such as Google Street View or OpenStreetMap (community-based 20 initiative) can be used. The usage of supporting documents such as existing local government land use plans and land cover maps, as well as actual verification of features at the ground level (ground truth, that is, information on surface features in the study area), is necessary, and was used in this study to verify land cover and land use features at the surface level.

Specific comments: The other reviewer provides excellent comments related to the technical correction that should be taken into account in the revision of the manuscript.

25 **Response:** The suggestions by anonymous referee #1 were largely taken into account (see previous section) for the editing of this manuscript.

(Attached is a copy of the revised manuscript with markup below).

**Spatial estimation of air PM2.5 emissions using activity data, local emission factors and land cover derived from satellite imagery**

Hezron P. Gibe1, Mylene G. Cayetano1

1Institute of Environmental Science and Meteorology, University of the Philippines, Diliman, Quezon City, Philippines

Correspondence to: Mylene G. Cayetano (mcayetano@iesm.upd.edu.ph)

Abstract. Exposure to air particulate matter (APM) is a serious environmental problem in many urban areas on Earth. presently relevant issue that affects the environment and the health of residents of many urban areas globally. In the Philippines, most existing studies and emission inventories have mainly focused on point and mobile sources, while research involving personal-human exposures to particulate pollutants is mostly lacking rare. This paper

- 10 presents a method for estimating the amount fine  $(PM_{2,5})$  particulate  $(PM_{2,5})$  emissions in a test study site in Cabanatuan City, Nueva Ecija in the Philippines, by utilizing local emission factors, regionally procured data and land cover/land use (activity data) interpreted from satellite imagery. The estimated emissions have been mapped using gGeographic information systems (GIS) software was used to map the estimated emissions in the study area. The present rResults suggest that vehicular emissions from motorcycles and tricycles, as well as biomass based
- 15 household fuels used by households (charcoal) and burning of agricultural waste largely contribute to PM2.5 emissions in Cabanatuan City. Overall, the method used in this study can be applied to any study site in other small urbanizing cities, as long as on-site specific activity data, emission factor and satellite-imaged land cover are available.

**Copyright statement**

5

20 This work is licensed under the Creative Commons Attribution 3.0 Unported License. To view a copy of this license, visit http://creativecommons.org/licenses/by/3.0/ or send a letter to Creative Commons, PO Box 1866, Mountain View, CA 94042, USA.

[revised manuscript text omitted]

(charcoal) Emission factors for vehicular activity (motorcycles/tricycles, jeepneys, PUVs) Emission factor for agricultural waste burning (rice straw)

In-house data

**2.3.2 Activity data**

The EEF for each emission type is calculated depending on which metrics are relevant for each source of PM2.5. Household and population data were obtained from local government documents, particularly the Comprehensive Land Use Plan(s) and Socio-Economic Profile(s) of Cabanatuan City; information on total amount of fuel used by household is obtained from the national Household Electricity Consumption Survey (HECS), conducted in 2005 and 2011. Table 21 compiles the sources of activity data used in this study2, in various units such as fuel consumption, population and household data, and agricultural land use data per year. Household and population data is obtained from local government documents, particularly the Comprehensive Land Use Plan(s) (CLUP) and Socio-Economic Profile(s) (SEP) of Cabanatuan City; information on total amount of fuel used by household is obtained from the national Household Electricity Consumption Survey (HECS), conducted in 2005 and 2011. Data on rice production as an indicator for agricultural waste production is obtained from the 2015 Cabanatuan City SEP. The findings of the study of Bakker, et al. (2013) is used as a reference to calculate how much agricultural waste (rice straw) is produced per amount of rice produced.

15

5

10

| Table 21. Data sources for activity data, childsfor factors, and Lix factor |
|------------------------------------------------------------------------------------|
|------------------------------------------------------------------------------------|

| Population data, land use                           | 2016 (provisional) Cabanatuan CLUP |
|-----------------------------------------------------|------------------------------------|
| •                                                   | Cabanatuan City SEP (2015)         |
| Activity data for households fuels; LPG,            | 2011/2005 Household Energy         |
| charcoal consumption                                | Consumption Survey (HECS)          |
| -                                                   | Ground surveys                     |
| Emission factors (PM 2.5 ) for fuelwood, | Cayetano and Lamorena (2014)       |
| charcoal                                            |                                    |
| Activity data for PUVs and motorcycles,             | Local government documents, Land   |
| tricycles (MC/TC),                                  | Transportation Office (LTO) annual |
|                                                     | reports, ground surveys            |
| Emission factors for PUVs and MC/TCs                | In-house data                      |
| Data on rice production and rice land               | Cabanatuan City SEP (2015)         |
| agricultural area                                   | 2016 (provisional) Cabanatuan CLUP |
| Data on rice straw generated per amount             | Bakker, et al. (2013)              |
| rice produced                                       |                                    |
| Emission factor for rice straw burning              | In house data                      |

**2.3.3 Emission estimation equations**

Emissions for household fuel (charcoal) were estimated with the formula shown in Eq. (2):

20  $E_{householdsfuels} = (N_h \times HF) \times Q_{fuel} \times EF \times 0.01$ , (2)

where:  $E_{\text{fuels}}$  is equal to emissions generated by charcoal fuels,  $N_h$  is the estimated number of households (generated from city government data), and HF is the factor (in percent)percentage of all households using charcoal as fuel, obtained from the HECS.  $Q_{\text{fuel}}$  is the quantity of fuel in kilograms used per year by each household, sourced from the HECS and verified using sensitivity analysis by ground surveys (see section 2.43). EF corresponds to the emission

factor for charcoal fuel PM25 <del>per year</del> per square kilometer per year; this is then multiplied by 0.01 to scale to each  $0.01 \text{ km}^2$  cell.

 $PM_{2.5}$  Expressions for motorcycles and tricycles vehicular sources were estimated with the formula shown in Eq. (3) and Eq. (4)<del>. (3)</del>.:

 $E_{vehiclesMC/TC} = (N_u \times DF \times NAVF) \times (EF \times KT \times SDF) \times 0.01$ , (3)

 $E_{PUV} = (N_u \times DF) \times EF \times 0.01$ (4)

where Factors that are the same for both equations include: Evenicles is equal to emissions generated by vehicles (motorcycles and tricycles, PUVs), Nu, is the estimated number of vehicle units, (by type: MC/TCs, PUVs); multiplied by density factor DF, the density factor corresponding to(-amount of vehicles per km2), and EF, the emission factor. area The in-house emission factor for MC/TCs is measured as  $PM_{2.5}$  per kilometer traveled (per

- 15 vehicle). Due to this non-standard EF unit, additional factors are required in Eq. (3). These include the in the city and non-association (vehicles) factor (AVNAF), the percentage of vehicles which are officially registered and properly accounted for by the citywhich corresponds to an additional multiplier to the overall number of vehicles taking into account unregistered vehicles (not registered by the city, or are from outside Cabanatuan City). To scale the EF to its proper units, it is multiplied by factor KT (kilometers traveled per day) and SDF (days in service per year). Similar to
- 20 the previous example, the total is also multiplied by 0.01 to scale to each 0.01 km2 cell. The DF and NAF were sourced and derived from city government data and was verified using sensitivity analysis by ground surveys as detailed in section 2.4 well. EF corresponds to the emission factor for motorcycle and tricycle/PUV PM2.5 per year per square kilometer per kilometer traveled. The emission factor is scaled to the average distance traveled by any given vehicle unit, here represented as factor KT (kilometers traveled). Similar to the previous example, also multiplied by 25 0.01 to scale to each 0.01 km2 cell.

Emissions for rice straw burning in agricultural areas agricultural waste burning were estimated with the formula shown in Eq. (54):

$$E_{strawagricultural} = \left(\frac{\text{RS}}{\text{RA}}\right) \times \text{EF} \times \text{SF} ,$$
(54)

where: Estraw is equal to emissions generated by rice straw burning, RS is the amount of rice straw produced per year, divided by RA, which is the total area in hectares (0.01 km2) used for growing of rice. EF is the in-house obtained emission factor for rice straw burning PM2.5 per year per square kilometer. SF is the survey factor, representing the percentage of farming area where burning of rice straw as agricultural waste is used. acting as Tthise reduction factor

35 is taken from the study of Launio, et al. (2013).; this represents the percentage of farming area where burning of rice straw as agricultural waste is used.

These equations are applied to estimate  $PM_{2.5}$  emissions for each cell, determined by its land cover type (households, vehicles, agricultural). After the estimated emissions for each cell have been calculated, they were mapped using

40 ArcMap (ArcGIS 10.1) software. For each emission source, aAll cells with estimated PM2.5 greater than zero assigned values (PM2.5 emissions above zero) were are plotted for each land cover type. according to the amount of PM2.5 
[revised manuscript text omitted]
 emissionsaccount for a small portion of vehicular emissions. PM2.5 from burning of <del>rice straw</del>agricultural waste accounts for a relatively was found to be a large portion constituent of total emissions-particulates within the study site\_., As the investigation area is only a small fraction of the entire city, this and is llikely means that
   agricultural waste burning is a significant source of PM2.5 in the largely agricultural Cabanatuan City. to account for emissions in agricultural zones of Cabanatuan City outside the study site; tThis is open to future research on air quality management in the city, among others.
- The equation used to estimate air particulate emissions produced a value The amount of PM2.5 emissions in the
  investigation area estimated by this method is comparable to emission levels in urban metropolitan areascities.,
  though tA possible reason for this is likely due to the common usage of biomass-based fuels such as charcoal, or the high levels of particulates from vehicular sourcesbeing responsible for more PM2.5 emissions as a major source.
  Vehicular emissions and agricultural waste burning, The estimated emission levels for tricycles and rice straw burning is of note, as these two factors at their highest levels, are responsible for have produced emission levels of at least 2

[revised manuscript text omitted]

---

## Author Response (AR3)

**Journal:** AMTD/AMT
**Title:** Spatial estimation of air PM$_{2.5}$ emissions using activity data, local emission factors and land cover derived from satellite imagery
**Author(s):** Hezron Gibe and Mylene Cayetano
5   **MS No.:** amt-2017-14

**Subject:** Authors' comment: point-by-point reply to comments by Associate Editor

We thank the Associate Editor for taking time to oversee the final review for this manuscript. The suggested edits for grammar and use of abbreviations were taken. References and citations to software, usage of satellite imagery, and data sources were also added. A few paragraphs were also included with additional information as suggested. We believe that these suggestions have
10   been important for increasing the quality of the text. We hope that this will be enough for your recommendation for this paper to be published to AMT.

Kindly refer to the following point-by-point replies to these comments. We appreciate your kind consideration and detailed comments for improving the content and preparing this manuscript for publication to the journal.

**Evaluation and response to interactive comment by Associate Editor:**

15   Dear Authors,

You have obviously made a great effort to improve the manuscript according to the referees' suggestions. Yet, I still have some comments that I would like to see addressed before the manuscript can be published. These are mostly about missing literature or internet references, the use of abbreviations, and some grammatical corrections. More importantly, I have two scientific questions: first, how can the results of your method be validated? Although your results look very plausible, validation of a new
20   method is crucial. And second, how useful would it be to provide seasonal PM2.5 maps? The contribution of rice straw burning, for example, clearly depends on the time of year.

Detailed comments:

1. Literature (or internet) references to Google Earth/Streetview, OpenStreetmap, and Arcmap / ArcGIS software are missing. Also an acknowledgment for the free use of data (Google and OpenStreetmap) is lacking.

25   **Response:** References were added to relevant points, particularly:

**Page 3, Line 24-28:** *Detailed images over the ground, taken by Google Street View (examples are shown in Fig. 3) was used to verify building types (residential/commercial). Satellite images were dated 3 March, 2016, while ground level (Street View) images were dated September 2015 (Google Earth Pro, 2015; Google Earth Pro, 2016). Additionally, maps from OpenStreetMap were also used for identifying special landmarks or as an additional resource since it occasionally presents*
30   *more updated information on surface features than Google Earth/Google Street View (OpenStreetMap, 2016).*

**Page 7, Line 8-10:** *These equations are applied to estimate PM2.5 emissions for each cell, determined by its land cover type (households, vehicles, agricultural). After the estimated emissions for each cell have been calculated, they were mapped using ArcMap (ArcGIS 10.1) software (ESRI, 2011). All cells with estimated PM2.5 greater than zero are plotted for each land cover type.*

35   An acknowledgement was added to its corresponding section:

**Page 1, Line 19-21:** *Map data used in this study is copyrighted (2015, 2016) to Google and data providers: Landsat, Copernicus, ZENRIN, and SKEnergy. Additional map data copyrighted to OpenStreetMap contributors and available from https://www.openstreetmap.org*

Their corresponding citations were also added to the references section.

40
* * *
Also other literature references, particularly to the activity data (Section 2.3.2) are missing. For example, cite the reference for the HECS as "Household Energy Consumption Survey (HECS)", Philippine Statistics Authority, https://psa.gov.ph/hecs, accessed on 7/7/2017.

**Response:** References were added to the following:

**Page 5, Line 12-16:** *Household and population data is obtained from local government documents, particularly the Comprehensive Land Use Plan(s) (CLUP) and Socio-Economic Profile(s) (SEP) of Cabanatuan City (Cabanatuan City CLUP, 2016; Cabanatuan City SEP, 2015). Information on total amount of fuel used by household is obtained from the national Household Electricity Consumption Survey (HECS), conducted in 2005 and 2011 (PSA, 2011).*

2. Please pay attention to the abbreviations you use, such as PUV, MC/TC, etc. These need to be explained at least once in the text, preferably when they are first introduced. Alternatively, you can add a table with abbreviations, but as the manuscript contains many tables already I would recommend the first option.

**Response:** Usage of these abbreviations have been improved for this paper, For all of these listed abbreviations, the sections in which they appear have now been re-written to introduce their meaning when they are first introduced, such as in:

**Page 2, Line 31-32:** *Sources corresponding to urban activity include vehicular mobile sources such as tricycles, jeepneys, and public utility vehicles (PUVs, which include buses and vans).*

**Page 6, Line 15-16:** *The in-house emission factor for motorcycles and tricycles (here abbreviated as MC/TCs) is measured as $PM_{2.5}$ per kilometer traveled (per vehicle).*

3. Can you comment on seasonality? Your study is done on a yearly basis, but rice straw is only burnt in certain seasons.

**Response:** A few sentences were added in various sections on seasonality of rice straw burning.

**Page 9, Line 12-17:** *The map of estimated emissions from rice straw burning is shown in Fig. 8. The amount of $PM_{2.5}$ here is assumed to represent the entire year, despite rice straw only being burned as agricultural waste in certain seasons. In particular, rice straw burning only occurs at the end of each planting season. This typically occurs around the months of April and October in Cabanatuan City.*

*Also, only a certain fraction of all agricultural land in the city is used in the growing of rice (this data is taken from the Cabanatuan City CLUP), and this was taken into account when estimating emissions for this map. [...]*

**Page 10, Line 30 – Page 11, Line 2:** *This method currently compiles estimated emissions on a yearly basis. The presence of seasonal factors such as agricultural waste burning, however, can indicate the possible usefulness of seasonal mapping of $PM_{2.5}$ emissions. Since it is equally important to investigate air pollution emissions through different temporal scales, such an option is worth looking into in the future.*

4. Please add a paragraph about how these data can be validated to Section 5. Satellite-based monitoring is probably not possible due to the small scale of the study region, so what methods would you suggest?

**Response:** A substantial edit was made to the latter half of Section 5, reordering paragraphs and adding content with regard to possible methods of validation of data.

**Page 11, Line 4-24:** *Additionally, a method for the verification of activity data factors, similar to this study's sensitivity analysis, is highly recommended for future studies. A focus on such studies but on a much larger scale, a ground survey that represents a much larger portion of an investigation area, would be instrumental in placing the total emission estimate more accurate with regards to specific conditions in a city. Actual in situ measurement of $PM_{2.5}$ emissions is also possible for a small study area like this one. Such a validation activity would require the use of air samplers or particle counters to actually measure the amount of $PM_{2.5}$ present. A limitation of this method, however, lies in the fact that it can only measure total particulate emissions, and cannot differentiate between different sources of $PM_{2.5}$. Because of this, any follow-up study that involves actual in situ $PM_{2.5}$ measurements must also include chemical analysis of sampled particulates as well as source apportionment in order to determine the actual amounts of air pollutants by source.*

*Lastly, as this method is primarily geared towards the estimation of particulate emissions, the planning of mitigation strategies to increase air quality in target cities such as in Cabanatuan City must also be pursued in tandem with emission inventories conducted by the local government and other stakeholders. Local governments in the Philippines are continuously upgrading its capabilities for spatial knowledge and city planning due to the propagation of usage of GIS software by government officials and non-government organizations (NGOs). This particular study has used ArcGIS, a proprietary software that requires a paid license, which may prove to be an issue for units with small financial capabilities. As this method can just as easily be executed using free and open source GIS software such as QGIS, studies using this software may be used in the future for organizations*

*seeking a less costly alternative for GIS. The specialization of city environment officers in pollution studies is a process that is both ongoing and needing more attention. For future studies and efforts, it will be worthwhile to increase the capability of local stakeholders to plan for environmental issues like air pollution.*

Line 6 - "air particulate matter (APM)" -> "particulate matter (PM)"

Line 8 - "...estimating the amount fine..." -> "...estimating the amount of fine..."

**Response:** These sentences have been corrected, as seen in:

**Page 1, Line 6-10:** *Exposure to particulate matter (PM) is a serious environmental problem in many urban areas on Earth. In the Philippines, most existing studies and emission inventories have mainly focused on point and mobile sources, while research involving human exposures to particulate pollutants is rare. This paper presents a method for estimating the amount of fine particulate ($PM_{2.5}$) emissions in a test study site in Cabanatuan City, Nueva Ecija in the Philippines, by utilizing local emission factors, regionally procured data and land cover/land use (activity data) interpreted from satellite imagery.*

Line 4 - "Sources of PM2.5 are caused by many man-made activities" -> "Enhancements in PM2.5 are mainly caused by various human activities"

Line 5 - "A common source of PM2.5..." -> "A common source of particles contributing to PM2.5..."

**Response:** These sentences have been corrected, as seen in:

**Page 2, Line 5-7:** *Enhancements in $PM_{2.5}$ are mainly caused by various human activities. A common source of particles contributing to $PM_{2.5}$, in urban areas is related to mobile sources, directly emitted by internal combustion processes inside vehicles of all types (Andrade, et al., 2012; Ahanchian and Biona, 2014; Chen, et al., 2016)*

Line 7 - "(such as CORINAIR and AP 42)" -> literature references are missing here

**Response:** References to emission factor guidebooks were added to this section:

**Page 2, Line 7-9:** *In most of the reports from Philippine cities, vehicular emissions reported in inventories use foreign emission factors (such as the 2007 version of the CORINAIR emission guidebook (EEA, 2007) and the Compilation of Air Pollutant Emission Factors (AP 42) (EPA, 1995)).*

Line 13 - "conducted" -> "constructed"

Line 16 - "sourced" -> "determined"

Line 18 - "which produces" -> "producing"

Line 26 - "This method is specifically meant to explore this method " -> "This study is specifically meant to explore this method "

**Response:** These sentences have been corrected, as seen in:

**Page 2, Line 14-15:** *Emission inventories in general have likewise not been constructed in many cities.*

**Page 2, Line 18-21:** *This study presents a method to estimate $PM_{2.5}$ by utilizing locally determined emission factors, satellite imagery, and activity data. The latter is obtained from interpretation of geographic information system (GIS) data and by identifying and localizing all sources in the city, taking into account the type of emission (point, area, mobile), and activities producing the emissions.*

**Page 2, Line 28-30:** *This study is specifically meant to explore this method for use in relatively small regional urban centers and cities in the Philippines; especially due to these cities being situated in locations where there is a mixture of rural and urban activities.*

Lines 17-19 - do you have literature references for this method?

**Response:** Sentences were added to the paragraph starting in Page 3, Line 19 to include a reference/influence for the method used in this study.

**Page 3, Line 19-23:** *The investigation area was divided with 24 x 40 grid cells (100 x 100 m or 1 ha / 0.01 km² each). For each cell, the type of man-made activity was interpreted from satellite images taken from Google Earth software. The classification process is similar to methods of supervised classification of land cover, as utilized by current local training activities on emission inventories such as the Clean Air for Smaller Cities (CASC) project (Yuberk and Cornet, 2013). The image of the surface feature is compared to a reference area of known land cover. Due to the size of each cell, the detail of each ground feature can be clearly seen. [...]*

Lines 26-28 - "(The collaged image used in Google Earth is sourced from processed images from Landsat and the European Space Agency (ESA)'s Copernicus program). "The Google Earth images used consist of post-processed Landsat images from the European Space Agency (ESA)'s Copernicus program."

**Response:** This sentence has been corrected, as seen in:

**Page 3, Line 30-31:** *Google Earth images have been used here instead of raw image data from, for example, the Landsat satellite. The Google Earth images used consist of post-processed Landsat images from the European Space Agency (ESA)'s Copernicus program.*

Line 2 - "Even so, " -> "Despite these disadvantages, "

**Response:** This sentence has been corrected, as seen in:

**Page 4, Line 6-7:** *Despite these disadvantages, the Google satellite image product is useful enough for the uninitiated considering the present purpose.*

Lines 8-15 (and accompanying Fig. 4) - Land cover type and expected dominating sources of PM are being mixed here. Please be more consistent, for example by changing the map to contain only the land type (i.e., substituting "fuels" by "traffic" and "grilling" by "market place" (What are commercial grilling establishments?) ). Then you can mention the assumed PM sources for each land use type in the text.

**Response:** The legend for the map shown in Figure 4 has been corrected to substitute certain terms: "Commercial", "Fuels", "Grilling", and "Residential" have been changed to "Commercial (Electricity)", "Residential (Charcoal)", "Commercial (Charcoal)", and "Residential (LPG)", respectively. "Fuels" here in this manuscript refers to household fuels, not vehicular fuels, and this first mention has been corrected to "charcoal" for clarity. The corresponding part of the manuscript has also been edited, as seen in the following:

**Page 4, Lines 15-20:** *Figure 4 shows that residential land use (cells marked as "Residential (LPG)"; households using liquefied petroleum gas as a fuel) are spread widely, although with noticeable commercial districts and open fields (not settled or occupied) located within this area. Two large agricultural areas are found in the northwest and east, occupied by small households likely using biomass-based fuels like charcoal (cells marked as "Residential (Charcoal)"). Another kind of commercial area is also indicated using cells marked as "Commercial (Charcoal)". These are areas with commercial establishments specializing in grilling foodstuffs highlighted as a possible specific source of $PM_{2.5}$ emissions.*

Line 28 - "sourced" -> "estimated"

**Response:** This sentence has been corrected, as seen in:

**Page 5, Line 4-5:** *Emission factors for households, vehicular emissions, and agricultural waste burning is estimated from various local studies and projects (Table 1).*

Line 15 - "GHGs" -> "greenhouse gases"

**Response:** This sentence has been corrected, as seen in:

**Page 10, Line 21-23:** *While this method was primarily developed to estimate PM$_{2.5}$, similar methods can be used for other components of the emission inventory process in the country (i.e. criteria pollutants and greenhouse gases).*

Figure 1. Change the caption to "Map of the Philippines and location of Cabanatuan City and some other major cities.", then add some major cities (or at least Manila) for reference.

**Response:** Figure 1 has been changed to add local cities for reference. The caption has also now been changed to read:

**Caption:** *Figure 1: Map of the Philippines and location of Cabanatuan City (with major cities)*

(Additional edits as of 7/29/17)

Formatted affiliations and added new text:

**Page 1, Line 3-6:** *Hezron P. Gibe[1] and Mylene G. Cayetano[1, 2] […] [2]International Environmental Research Institute, Gwangju Institute of Science and Technology, Cheomdan-gwagiro, Buk-gu, 500-712 Gwangju, South Korea*

Minor edit:

**Page 1, Line 21-22:** *Map data (satellite and street-level imagery) used in this study is copyrighted (2015, 2016) to Google and data providers: Landsat, Copernicus, ZENRIN, and SKEnergy.*

Added new citation for Ahanchian and Biona (2014)

**Page 12, Line 6-9:** *Ahanchian, M., Biona, J., Energy demand, emissions forecasts and mitigation strategies modeled over a medium-range horizon: the case of the land transportation sector in Metro Manila. Energ. Policy. 66. 615-629. doi: 10.1016/j.enpol.2013.11.026. 2014.*

Attached below is the text of the manuscript with markup:

**Spatial estimation of air PM$_{2.5}$ emissions using activity data, local emission factors and land cover derived from satellite imagery**

Hezron P. Gibe[1], and Mylene G. Cayetano[1, 2]

[1]Institute of Environmental Science and Meteorology, University of the Philippines, Diliman, Quezon City, Philippines
[2]International Environmental Research Institute, Gwangju Institute of Science and Technology, Cheomdan-gwagiro, Buk-gu, 500-712 Gwangju, South Korea

*Correspondence to*: Mylene G. Cayetano (mcayetano@iesm.upd.edu.ph)

**Abstract.** Exposure to air particulate matter (APM) is a serious environmental problem in many urban areas on Earth. In the Philippines, most existing studies and emission inventories have mainly focused on point and mobile sources, while research involving human exposures to particulate pollutants is rare. This paper presents a method for estimating the amount of fine particulate (PM$_{2.5}$) emissions in a test study site in Cabanatuan City, Nueva Ecija in the Philippines, by utilizing local emission factors, regionally procured data and land cover/land use (activity data) interpreted from satellite imagery. Geographic information system (GIS) software was used to map the estimated emissions in the study area. The present results suggest that vehicular emissions from motorcycles and tricycles, as well as fuels used by households (charcoal) and burning of agricultural waste largely contribute to PM$_{2.5}$ emissions in Cabanatuan City. Overall, the method used in this study can be applied in other small urbanizing cities, as long as on-site specific activity data, emission factor and satellite-imaged land cover are available.

**Copyright statement**

This work is licensed under the Creative Commons Attribution 3.0 Unported License. To view a copy of this license, visit http://creativecommons.org/licenses/by/3.0/ or send a letter to Creative Commons, PO Box 1866, Mountain View, CA 94042, USA. Map data (satellite and street-level imagery) used in this study is copyrighted (2015, 2016) to Google and data providers: Landsat, Copernicus, ZENRIN, and SKEnergy. Additional map data copyrighted to OpenStreetMap contributors and available from https://www.openstreetmap.org

[revised manuscript text omitted]

**Roads**

⎯⎯⎯ Arterial roads

⎯⎯⎯ Collector roads

⎯⎯⎯ Residential roads

⎯⎯⎯ Unclassified minor roads

☐ Poblacion barangays

☐ Barangays

**Cabanatuan Grid**

**Land cover type**

■ Agricultural

■ Cemetery

■ Commercial

■ Fuels

■ Grilling

■ Residential

☐ Open

■ River

■ Terminal

N

0  0.2 0.4    0.8     1.2     1.6     2
⌐Kilometers

[Figure]

**Figure 4: Land cover/land use map from interpretation of satellite image.**

[Figure]

**Figure 5: Map of estimated PM₂.₅ emissions from burning of household fuels.**

[Figure]

**Figure 6: Map of estimated PM₂.₅ emissions from motorcycles and tricycles.**

[Figure]

**Figure 7: Map of estimated PM₂.₅ emissions from PUVs (public utility vehicles/jeepneys/XLTs).**

[Figure]

**Figure 8: Map of estimated PM₂.₅ emissions from burning of rice straw as agricultural waste.**

[Figure]

**Figure 9: Map of estimated PM₂.₅ emissions combining all factors in the study.**